# Infant brain regional cerebral blood flow increases supporting emergence of the default-mode network

**Qinlin Yu[1,2], Minhui Ouyang[1,2], John Detre[2,3], Huiying Kang[1,4], Di Hu[1,4], Bo Hong[5], Fang Fang[6], Yun Peng[4], Hao Huang[1,2]\***

[1]Department of Radiology, Children's Hospital of Philadelphia, Philadelphia, United States; [2]Department of Radiology, Perelman School of Medicine, University of Pennsylvania, Philadelphia, United States; [3]Department of Neurology, Perelman School of Medicine, University of Pennsylvania, Philadelphia, United States; [4]Department of Radiology, Beijing Children's Hospital, Capital Medical University, Beijing, China; [5]Department of Biomedical Engineering, Tsinghua University, Beijing, China; [6]School of Psychological and Cognitive Sciences, Peking University, Beijing, China

**Abstract** Human infancy is characterized by most rapid regional cerebral blood flow (rCBF) increases across lifespan and emergence of a fundamental brain system default-mode network (DMN). However, how infant rCBF changes spatiotemporally across the brain and how the rCBF increase supports emergence of functional networks such as DMN remains unknown. Here, by acquiring cutting-edge multi-modal MRI including pseudo-continuous arterial-spin-labeled perfusion MRI and resting-state functional MRI of 48 infants cross-sectionally, we elucidated unprecedented 4D spatiotemporal infant rCBF framework and region-specific physiology–function coupling across infancy. We found that faster rCBF increases in the DMN than visual and sensorimotor networks. We also found strongly coupled increases of rCBF and network strength specifically in the DMN, suggesting faster local blood flow increase to meet extraneuronal metabolic demands in the DMN maturation. These results offer insights into the physiological mechanism of brain functional network emergence and have important implications in altered network maturation in brain disorders.

**\*For correspondence:**
huangh6@email.chop.edu

**Competing interest:** The authors declare that no competing interests exist.

## Editor's evaluation

In this paper, the authors find a link between the emergence of functional connectivity (FC) and changes in regional Cerebral Blood Flow (rCBF) in human infancy from birth to 24 months of age, which will be of interest to the increasing field investigating how the establishment of the brain's functional organization is linked to neurodevelopmental and psychiatric conditions. The data quality and complementarity are impressive for infants over this developmental period (0-2 years). Most of the key claims of the manuscript are well supported by the data. However, the relatively sparse sample and cross-sectional nature do limit interpretation.

## Introduction

The adult human brain receives 15–20% of cardiac output despite only representing 2% of body mass (*Bouma and Muizelaar, 1990*; *Satterthwaite et al., 2014*). Vast energy demand from the human brain starts from infancy, which is characterized by fastest energy expenditure increase across lifespan (*Pontzer et al., 2021*). Infancy is also the most dynamic phase of brain development across entire

lifespan with fastest functional and structural brain development. For example, during infancy the brain size increases dramatically in parallel with rapid elaboration of new synapses, reaching 80–90% of lifetime maximum by age of year 2 (*Knickmeyer et al., 2008*; *Ouyang et al., 2019a*; *Pfefferbaum et al., 1994*). Structural and functional changes of infant brain are underlaid by rapid and precisely regulated (*Huang et al., 2013*; *Silbereis et al., 2016*) spatiotemporal cellular and molecular processes, including neurogenesis and neuronal migration (*Rakic, 1995*; *Sidman and Rakic, 1973*), synaptic formation (*Huttenlocher and Dabholkar, 1997*), dendritic arborization (*Bystron et al., 2008*; *Ouyang et al., 2019b*), axonal growth (*Haynes et al., 2005*; *Innocenti and Price, 2005*), and myelination (*Miller et al., 2012*; *Yakovlev, 1967*). These developmental processes demand rapidly increasing energy consumption of the brain. However, there have been few whole-brain mappings of heterogeneous infant brain regional cerebral blood flow (rCBF) changes across landmark infant ages from 0 to 24 months thus far, impeding understanding of energy expenditure across functional systems of early developing brain. As a result of differential neuronal growth across cortex, functional networks in the human brain develop differentially following the order from primary sensorimotor to higher-order cognitive systems (*Cao et al., 2017a*; *Huang and Vasung, 2014*; *Sidman and Rakic, 1982*; *Tau and Peterson, 2010*; *Yu et al., 2016*). The default-mode network (DMN) (*Raichle et al., 2001*) is widely recognized as a fundamental neurobiological system associated with cognitive processes that are directed toward the self and has important implication in typical and atypical brain development (*Buckner et al., 2008*). Unlike primary sensorimotor (SM) and visual (Vis) networks emerging relatively earlier around and before birth (*Cao et al., 2017b*; *Doria et al., 2010*; *Smyser et al., 2010*), emergence of the vital resting-state DMN is not well established until late infancy (*Gao et al., 2009*). Till date, it has been unclear how emergence of vital functional networks such as DMN is coupled with rCBF increase during infancy.

Regional brain metabolism, including glucose utilization and oxygen consumption, is closely coupled to regional CBF (rCBF) that delivers the glucose and oxygen needed to sustain metabolic needs (*Raichle et al., 2001*; *Vaishnavi et al., 2010*). Infant rCBF has been conventionally measured with positron emission tomography (PET) (*Altman et al., 1988*; *Altman et al., 1993*; *Chugani and Phelps, 1986*; *Chugani et al., 1987*) and single-photon emission computerized tomography (SPECT) (*Chiron et al., 1992*), which are not applicable to infants due to the associated exposure to radioactive tracers. By labeling the blood in internal carotid and vertebral arteries in neck and measuring downstream labeled arterial blood in brain, arterial-spin-labeled (ASL) (*Alsop et al., 2015*; *Detre and Alsop, 1999*) perfusion MRI provides a method for noninvasive quantifying rCBF without requiring radioactive tracers or exogenous contrast agents. Accordingly, ASL is especially suitable for rCBF measurements of infants (*Ouyang et al., 2017*; *Lemaître et al., 2021*; *Wang et al., 2008*) and children (*Jain et al., 2012*; *Satterthwaite et al., 2014*). Phase-contrast (PC) MRI, utilizing the phase shift proportional to velocity of the blood spins, has also been used to measure global CBF of the entire brain (*Liu et al., 2019*). Through integration of pseudo-continuous ASL (pCASL) and PC MRI, rCBF measured from pCASL can be calibrated by global CBF from PC MRI for more accurate infant brain rCBF measurement (*Aslan et al., 2010*; *Ouyang et al., 2017*). With rCBF closely related to regional cerebral metabolic rate of oxygen ($CMRO_2$) and glucose (CMRGlu) at the resting state in human brain (*Fox and Raichle, 1986*; *Gur et al., 2009*; *Paulson et al., 2010*; *Vaishnavi et al., 2010*), rCBF could be used as a surrogate measure of local cerebral metabolic level for resting infant brains.

Early developing human brain functional networks can be reproducibly measured with resting-state fMRI (rs-fMRI). For example, a large scale of functional architecture at birth (*Cao et al., 2017b*; *Doria et al., 2010*; *Fransson et al., 2007*) has been revealed with rs-fMRI. Functional networks consist of densely linked hub regions to support efficient neuronal signaling and communication. These hub regions can be delineated with data-driven independent component analysis (ICA) of rs-fMRI data and serve as functional regions of interest (ROIs) for testing physiology–function relationship. Meeting metabolic demands in these hub ROIs is critical for functional network maturation. In fact, spatial correlation of rCBF to the functional connectivity (FC) in these functional ROIs was found in adult brains (*Liang et al., 2013*). Altered DMN plays a vital role in neurodevelopmental disorders such as autism (*Doyle-Thomas et al., 2015*; *Lynch et al., 2013*; *Padmanabhan et al., 2017*; *Washington et al., 2014*). Thus, understanding physiological underpinning of the DMN maturation offers invaluable insights into the mechanism of typical and atypical brain development.

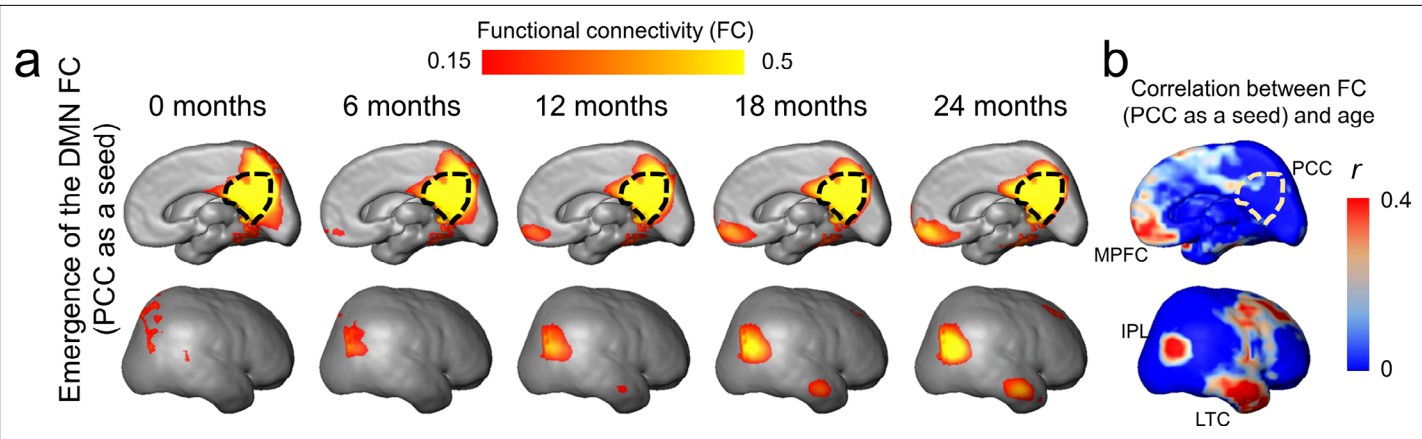

**Figure 1.** Emergence of functional connectivity (FC) within the default-mode network (DMN) during infancy. The maps of the DMN FC (PCC as a seed) at representative ages from 0 to 24 months are demonstrated in (**a**), and the map of correlation coefficient of FC (PCC as a seed) and age is demonstrated in (**b**). In (**a**), gradually emerging FC of other DMN regions (including MPFC, IPL, and LTC) to the PCC from 0 to 24 months can be appreciated. The PCC is delineated by the black dashed contour. In (**b**), stronger correlation between FC (PCC as a seed) and age is localized in DMN subregions IPL, ITL, and MPFC. Abbreviations of DMN subregions: IPL: inferior posterior lobule; LTC: lateral temporal cortex; MPFC: medial prefrontal cortex; PCC: posterior cingulate cortex.

The online version of this article includes the following figure supplement(s) for figure 1:

**Figure supplement 1.** Identification of functional network regions of interest with resting-state fMRI of infants aged 12–24 months.

**Figure supplement 2.** Age-dependent changes of functional connectivity (FC) within the default-mode network (DMN), visual (Vis), and sensorimotor (SM) network regions during infancy.

We hypothesized heterogeneous rCBF maps at landmark infant ages and faster rCBF increase in brain regions of higher cognitive functions (namely DMN regions) during infancy than those of primary sensorimotor functions where functional networks emerge before or around birth (*Cao et al., 2017a*; *Cao et al., 2017b*; *Doria et al., 2010*; *Fransson et al., 2007*; *Smyser et al., 2010*; *Peng et al., 2020*). Furthermore, with rCBF as an indicator of local metabolic level of glucose and oxygen consumption, we hypothesized that strongly coupled rCBF and FC increase specifically in the DMN regions during infancy to meet extra metabolic demand of DMN maturation. In this study, we acquired multi-modal MRI, including both pCASL perfusion MRI, and rs-fMRI, of 48 infants aged 0–24 months to quantify rCBF and FC, respectively. RCBF at the voxel level and in functional network ROIs were measured to test the hypothesis of spatiotemporally differential rCBF increases during infancy. Maturation of FC in the DMN was delineated. Correlation of FC increase and rCBF increase in the DMN ROIs was tested and further confirmed with data-driven permutation analysis, the latter of which was to examine whether the coupling of rCBF and FC takes place only in the DMN during infancy.

## Results

### Emergence of the DMN during early brain development

*Figure 1a* shows the emergence of the DMN in typically developing brain from 0 to 24 months as measured using rs-fMRI with a posterior cingulate cortex (PCC, a vital hub of DMN network) seed region indicated by the black dash line. At around birth (0 months), the DMN is still immature with weak FC between PCC and other DMN regions, including medial prefrontal cortex (MPFC), inferior posterior lobule (IPL), and lateral temporal cortex (LTC) (*Figure 1a*). During infant brain development from 0 to 24 months, *Figure 1a* shows that the functional connectivity between MPFC, IPL, or ITC and PCC gradually strengthens. *Figure 1b* shows the FC–age correlation *r* value map. It can be appreciated from *Figure 1b* that across the cortical surface relatively higher r values are only located at the DMN regions (except the seed PCC).

For robust and consistent identification of functional network ROI of infants, three functional network ROIs, including DMN, visual (Vis) network, and sensorimotor (SM) network, were generated from rs-fMRI data of infant aged 12–24 months, as shown in *Figure 1—figure supplement 1*. After

applying these network ROIs to measure functional connectivity changes of infants of all ages from 0 to 24 months, we found significant increase of the FC only within the DMN ($r = 0.31$, p<0.05), but not in the Vis ($r = 0.048$, p=0.745) or SM ($r = 0.087$, p=0.559), network regions (*Figure 1—figure supplement 2*), indicating significant functional development in the DMN, but not in the Vis or SM network.

## Faster rCBF increases in the DMN hub regions during infant brain development

The labeling plane and imaging slices of pCASL perfusion MRI of a representative infant brain, reconstructed internal carotid and vertebral arteries, and four PC MR images of the target arteries are shown in *Figure 2a*. The rCBF maps of infant brains were calculated based on pCASL perfusion MRI and calibrated by PC MRI. As an overview, axial rCBF maps of typically developing brains at milestone ages of 1, 6, 12, 18, and 24 months are demonstrated in *Figure 2b*. The rCBF maps with high gray/white matter contrasts can be appreciated by a clear contrast between white matter and gray matter. A general increase of blood flow across the brain gray matter from birth to 2 years of age is readily observed. Heterogeneous rCBF distribution at a given infant age can be appreciated from these maps. For example, higher rCBF values in primary visual cortex compared to other brain regions are clear in younger infant at around 1 month. *Figure 2b* also demonstrates differential rCBF increases across brain regions. RCBF increases are prominent in the PCC, indicated by green arrows. On the other hand, rCBF in the visual cortex is already higher (indicated by blue arrows) than other brain regions in early infancy and increases slowly across infant development. The adopted pCASL protocol is highly reproducible with intraclass correlation coefficient (ICC) 0.8854 calculated from pCASL scans of a randomly selected infant subject aged 17.6 month, shown in *Figure 2—figure supplement 1*. With rCBF measured at these functional network ROIs, *Figure 2—figure supplement 2* quantitatively exhibits spatial inhomogeneity of rCBF distribution regardless of age. These quantitative measurements are consistent to the observation of heterogeneous rCBF distribution in *Figure 2b*. Specifically, as shown in *Figure 2—figure supplement 2*, significant heterogeneity of rCBF was found across regions ($F(6, 282) = 122.6$, p<$10^{-10}$) with an analysis of variance (ANOVA) test. With further paired *t*-test between regions, the highest and lowest rCBF was found in the Vis ($82.1 \pm 2.19$ ml/100 g/min) and SM ($49.1 \pm 1.49$ ml/100 g/min) regions, respectively (all $ts(47) > 4.17$, p<0.05, false discovery rate [FDR] corrected), while rCBF in the DMN ($67.8 \pm 2.08$ ml/100 g/min) regions was in the middle (all $ts(47) > 2.87$, p<0.05, FDR corrected). Within the DMN, rCBF in the PCC ($75.4 \pm 2.19$ ml/100 g/min) and LTC ($72.0 \pm 2.82$ ml/100 g/min) regions were significantly higher than rCBF in the MPFC ($60.7 \pm 2.24$ ml/100 g/min) (both $ts(47) > 8.22$, p<0.05, FDR corrected) and IPL regions ($59.4 \pm 1.96$ ml/100 g/min) (both $ts(47) > 7.87$, p<0.05, FDR corrected). After comparing corresponding rCBF measures of different network ROIs between left and right hemisphere for evaluating rCBF asymmetry, we found significantly higher ($ts(47) = 3.82$, p<0.05) rCBF in the SM network ROI in the right hemisphere ($50.8 \pm 1.67$ ml/100 g/min) compared to that in the left hemisphere ($47.8 \pm 1.43$ ml/100 g/min) while no significant rCBF difference was found in the DMN or Vis network ROIs between two hemispheres. This finding of rCBF asymmetry in the SM network ROI is consistent to the previous studies (*Chiron et al., 1997*; *Lemaître et al., 2021*).

Figure 3a shows cortical maps of linearly fitted rCBF values of infant brains from 0 to 24 months. Consistent with nonuniform profile of the rCBF maps observed in *Figure 2b*, the three-dimensionally reconstructed rCBF distribution maps in *Figure 3a* are also not uniform at each milestone infant age. RCBF increases from 0 to 24 months across cortical regions are apparent, as demonstrated by the relatively high rCBF–age correlation r values across the cortical surface in *Figure 3b*. Heterogeneity of rCBF increases across all brain voxels can be more clearly appreciated in *Figure 3a and b* compared to *Figure 2b*. Significant interaction between regions and age was found ($F(6, 322) = 2.45$, p<0.05) with an analysis of covariance (ANCOVA) test where age was used as a covariate. With DMN functional network regions including PCC, MPFC, IPL, and LTC as well as Vis and SM network regions delineated in *Figure 1—figure supplement 1b* as ROIs, rCBF trajectories in *Figure 3c* demonstrate that rCBF in these ROIs all increase significantly with age (Vis: $r = 0.53$, p<$10^{-4}$; SM: $r = 0.52$, p<$10^{-4}$; DMN: $r = 0.7$, p<$10^{-7}$; DMN_PCC: $r = 0.66$, p<$10^{-6}$; DMN_MPFC: $r = 0.67$, p<$10^{-6}$; DMN_IPL: $r = 0.66$, p<$10^{-6}$; DMN_LTC: $r = 0.72$, p<$10^{-8}$). Using the trajectory of primary sensorimotor (SM) (black line and circles) in *Figure 3c* as a reference, rCBF increase rates across functional network ROIs are also heterogeneous (*Figure 3c*). Specifically, significantly higher (all p<0.05, FDR corrected) rCBF increase rate was

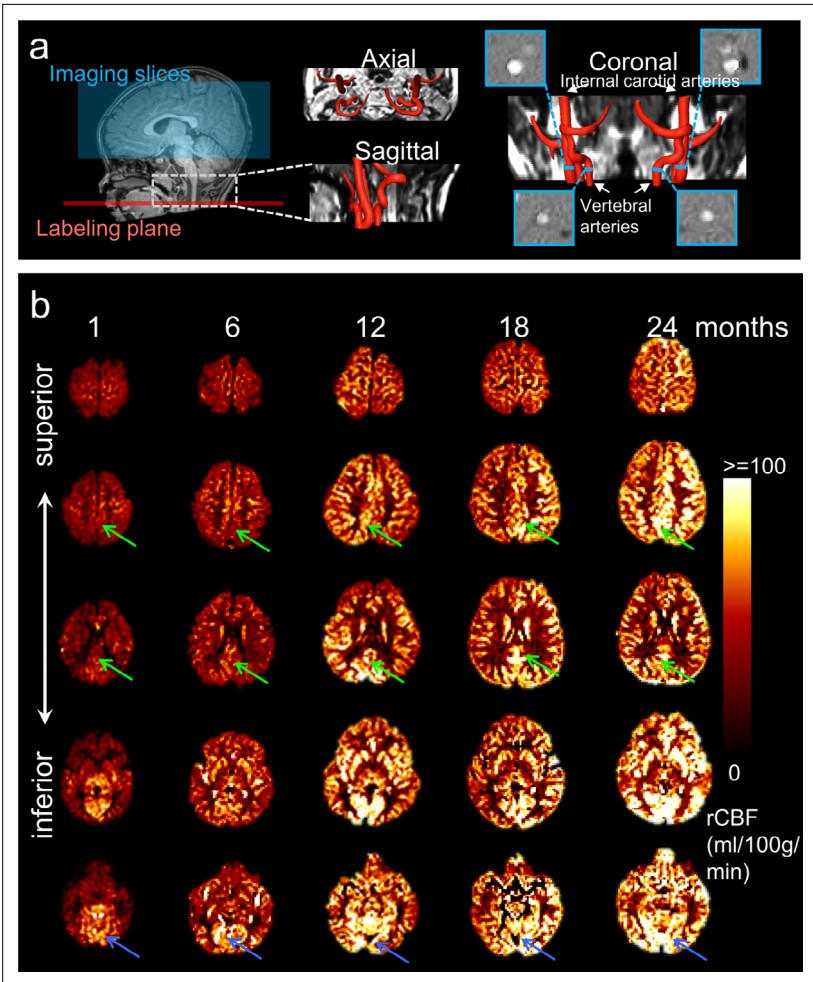

**Figure 2.** Acquisition of high-quality infant pseudo-continuous arterial-spin-labeled (pCASL) perfusion and phase contrast (PC) MRI and resultant axial regional cerebral blood flow (rCBF) maps at different infant ages. (**a**) Labeling plane (red line) and imaging volume (blue box) of pCASL perfusion MRI are shown on the mid-sagittal slice of T1-weighted image of a representative infant on the left panels. Axial and sagittal view of MR angiography with reconstructed internal carotid and vertebral arteries are shown in the middle of panel (**a**). On the right of panel (**a**), the coronal view of the reconstructed arteries is placed in the middle with four slices (shown as blue bars) of the PC MR scans positioned perpendicular to the respective feeding arteries. The PC MR images are shown on the four panels surrounding the coronal view of the angiography. These PC MR images measure the global cerebral blood flow of internal carotid and vertebral arteries and are used to calibrate rCBF. (**b**) rCBF maps of representative typically developing (TD) infant brains at 1, 6, 12, 18, and 24 months from left to right. Axial slices of rCBF maps from inferior to superior are shown from bottom to top of the panel b for each TD infant brain. Green arrows point to the posterior cingulate cortex (a hub of the DMN network) characterized by relatively lower rCBF at early infancy and prominent rCBF increases from 1 to 24 months. Blue arrows point to the visual cortex characterized by relatively higher rCBF at early infancy and relatively mild rCBF increase from 1 to 24 months.

The online version of this article includes the following figure supplement(s) for figure 2:

**Figure supplement 1.** Highly reproducible pseudo-continuous arterial-spin-labeled (pCASL) protocol adopted in this study for measuring regional cerebral blood flow (rCBF).

**Figure supplement 2.** Heterogeneity of regional cerebral blood flow (rCBF) measurements across functional network regions.

found in total DMN ROIs (1.59 ml/100 g/min/month) and individual DMN ROIs, including DMN_PCC (1.57 ml/100 g/min/month), DMN_MPFC (1.63 ml/100 g/min/month), DMN_IPL (1.42 ml/100 g/min/month), and DMN_LTC (2.22 ml/100 g/min/month) compared to in the SM ROI (0.85 ml/100 g/min/month). Although the rCBF growth rate in the Vis (1.27 ml/100 g/min/month) ROIs is higher than that

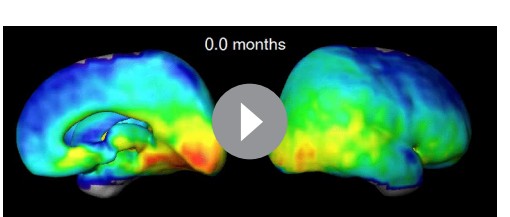

**Figure 3.** 4D spatiotemporal regional cerebral blood flow (rCBF) dynamics and faster rCBF increases in the default-mode network (DMN) hub regions during infancy. (**a**) Medial (top row) and lateral (bottom row) views of fitted rCBF profiles of the infant brain at 0, 6, 12, 18, and 24 months in the custom-made infant template space demonstrate heterogeneous rCBF increase across the brain regions. (**b**) Medial (top) and lateral (bottom) views of rCBF–age correlation coefficient (r) map are demonstrated. (**c**) The scatterplots of rCBF measurements in the primary sensorimotor (SM) network (black circle and black line), visual (Vis) network (blue circle and blue line), and total and individual DMN hub regions (DMN_MPFC, DMN_PCC, DMN_IPL, and DMN_LTC) (blue circle and blue line) of all studied infants demonstrate differential rCBF increase rates. * next to network name in each plot indicates significant (false discovery rate [FDR]-corrected p<0.05) differences of rCBF trajectory slopes from that of SM used as a reference and shown in a black dashed line. See legend of *Figure 1* for abbreviations of the DMN subregions.

in the SM ROIs, this difference is not significant (p=0.13). Collectively, *Figures 2 and 3* show that the CBF increases significantly and differentially across brain regions during infancy, with rCBF in the DMN hub regions increasing faster than rCBF in the SM and Vis regions (*Figure 3*). The 4D spatiotemporal whole-brain rCBF changes during infant development are presented in *Video 1*.

**Video 1.** Video of the 4D spatiotemporal whole-brain dynamics of regional cerebral blood flow from 0 to 24 months.

https://elifesciences.org/articles/78397/figures#video1

## Coupling between rCBF and FC within DMN during infant brain development

To test the hypothesis that rCBF increases in the DMN regions underlie emergence of this vital functional network, correlation between rCBF and FC was conducted across randomly selected voxels within the DMN of all infants aged 0–12 months (*Figure 4a*) and all infants aged 12–24 months (*Figure 4b*). Significant correlations (p<0.001)

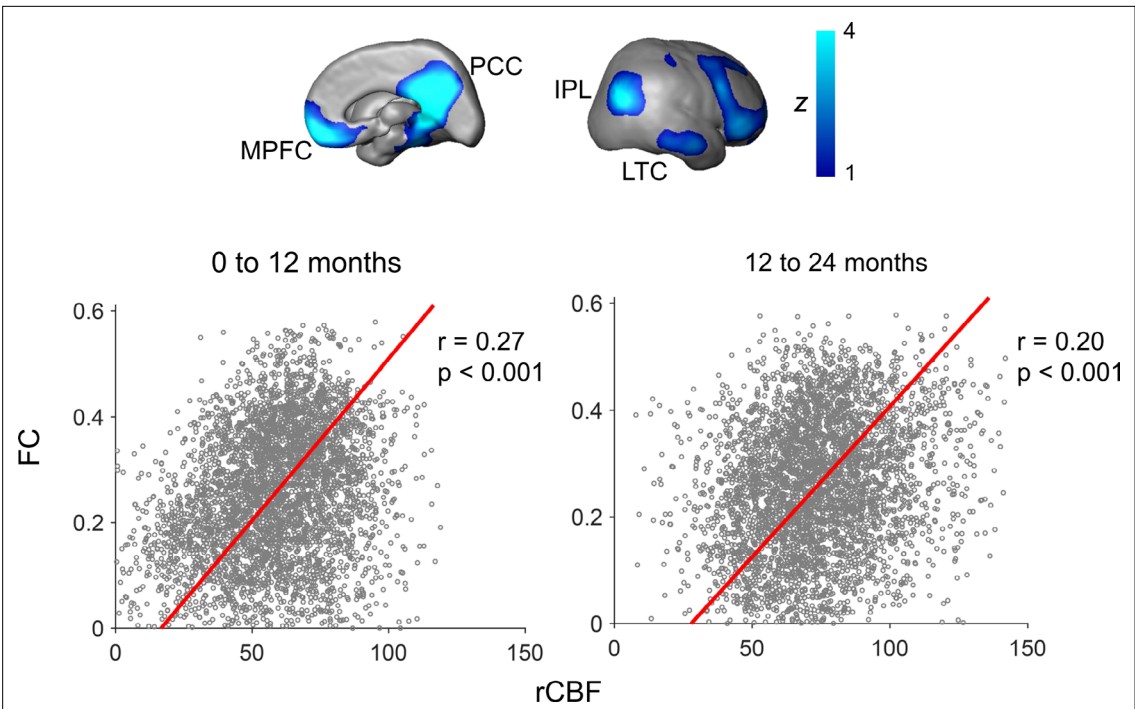

**Figure 4.** Significant correlation of regional cerebral blood flow (rCBF) and functional connectivity (FC) at randomly selected 4000 voxels within the default-mode network (DMN) for both infants aged 0–12 months (p<0.001, left scatter plot) and infants aged 12–24 months (p<0.001, right scatter plot). FC is the average of FC of a certain DMN voxel to all other DMN voxels. The DMN regions of interests obtained from a data-driven independent component analysis of resting-state fMRI of the 12–24month infant cohort are shown on the top panels as an anatomical reference. See legend of *Figure 1* for abbreviations of the DMN subregions.

were found in both age groups. Partial correlation analysis between rCBF and FC after regressing out age effects also confirmed significant correlation between rCBF and FC in the DMN regions in both 0–12-month (p<0.001) and 12–24-month (p<0.001) groups excluding the age effect. We further tested whether functional emergence of the DMN represented by increases of FC within the DMN (namely DMN FC) was correlated to rCBF increases specifically in the DMN regions, but not in primary sensorimotor (Vis or SM) regions. *Figure 5a* shows correlations between the DMN FC and rCBF at the DMN (red lines), Vis (green lines), or SM (blue lines) voxels. The correlations between the DMN FC and averaged rCBF in the DMN, Vis, or SM region are represented by thickened lines in *Figure 5a*. A correlation map (*Figure 5b*) between the DMN FC and rCBF across the entire brain voxels was generated. The procedures of generating this correlation map are illustrated in *Figure 5—figure supplement 1*. The DMN, Vis, and SM ROIs in *Figure 5b* were delineated with dashed red, green, and blue contours, respectively, and obtained from *Figure 1—figure supplement 1b*. Most of the significant correlations ($r > r_{crit}$) between the DMN FC and voxel-wise rCBF were found in the voxels in the DMN regions, such as PCC, IPL, and LTC, but not in the Vis or SM regions (*Figure 5b*). Demonstrated in a radar plot in *Figure 5c*, much higher percent of voxel with significant correlations between rCBF and the DMN FC was found in the DMN (36.7%, p<0.0001) regions than in the SM (14.6%, p>0.05) or Vis (5.5%, p>0.05) regions. Statistical significance of higher percent of voxels with significant correlations in the DMN (p<0.0001) was confirmed using nonparametric permutation tests with 10,000 permutations. We also conducted the correlation between the Vis FC and rCBF across the brain as well as permutation test. As expected, no significant correlation between the Vis FC and rCBF can be found in any voxel in the DMN, Vis, or SM ROIs, demonstrated in *Figure 5—figure supplement 2a*. Similar analysis was also conducted for correlation between the SM FC and rCBF across the brain and percent of voxels with significant correlation was close to zero, as demonstrated in *Figure 5—figure supplement 2b*. Combined with the results shown in *Figure 5*, the results of coupling between Vis (*Figure 5—figure supplement 2a*) or SM (*Figure 5—figure supplement 2b*) FC and rCBF further

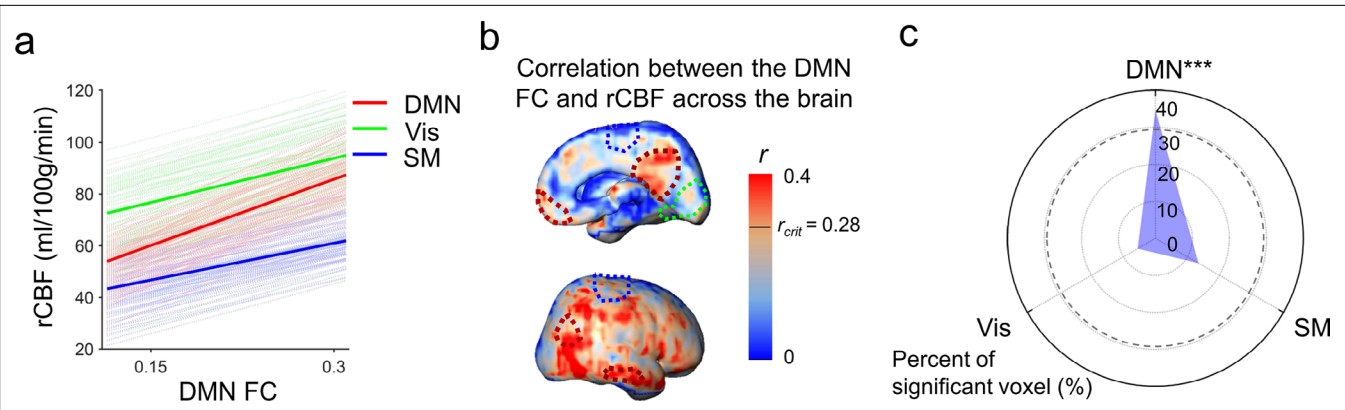

**Figure 5.** Significant correlation between functional emergence of the default-mode network (DMN) and regional cerebral blood flow (rCBF) increases specifically in the DMN regions, but not in primary sensorimotor (visual or sensorimotor) regions. (**a**) Correlation of intra-default-mode-network functional connectivity (DMN FC) and rCBF at randomly selected voxels in the DMN (light red lines), visual (Vis, light green lines) and sensorimotor (SM, light blue lines) network regions. Correlations of DMN FC and averaged rCBF in the DMN, Vis, and SM network regions are shown as thickened red, green, and blue lines, respectively. (**b**) Coupling between the DMN FC and rCBF across the brain can be appreciated by distribution of voxel-wise correlation coefficient (r) obtained from correlation between DMN FC and rCBF at each voxel. The short black line in the color bar indicates critical r value $r_{crit}$ corresponding to p=0.05. Higher r values can be appreciated in the DMN hub regions including posterior cingulate cortex (PCC), medial prefrontal cortex (MPFC), inferior posterior lobule (IPL), and lateral temporal cortex (LTC) with their boundaries delineated by the dashed dark red contours (from *Figure 1—figure supplement 1b*). Dashed green and blue contours (also from *Figure 1—figure supplement 1b*) delineate the Vis and SM network regions, respectively. (**c**) Radar plot shows significant correlation between rCBF and intra-DMN FC in the DMN network (36.7%, p<0.0001), but not in the Vis (14.6%, p>0.05), or SM (5.5%, p>0.05) networks. The radius represents the percent of the voxels with significant correlations between intra-DMN FC and rCBF in DMN, Vis, and SM network regions, respectively. The dashed line circle indicates critical percent of significant voxels with p=0.05 from 10,000 permutation tests. ***p<0.0001.

The online version of this article includes the following figure supplement(s) for figure 5:

**Figure supplement 1.** Procedures of generating map of correlation between the default-mode network (DMN) functional connectivity (FC) and reginal cerebral blood flow (rCBF) across the entire brain voxels.

**Figure supplement 2.** Coupling between intra-visual network functional connectivity (Vis FC) and regional cerebral blood flow (rCBF) and coupling between intra-sensorimotor network (SM FC) and rCBF.

**Figure supplement 3.** A diagram illustrating hypothesized neuronal mechanism supporting coupling of regional cerebral blood flow (rCBF) and functional connectivity.

demonstrated that the selected rCBF-FC coupling can be only found in the DMN ROIs, but not in the Vis or SM network ROIs.

## Discussion

We revealed strongly coupled rCBF and FC increases specifically in the DMN while establishing unprecedented 4D rCBF spatiotemporal changes during infancy. The tight rCBF-FC relationship found with multimodal infant MRI suggests that DMN emergence is supported by faster local blood flow increase in the DMN to meet metabolic demand, offering refreshing insight into the physiological mechanism underlying early brain functional architecture emergence. The delineated 4D brain perfusion spatiotemporal framework was characterized with heterogeneous rCBF distribution across brain regions at a specific age and differential age-dependent rCBF increase rates across brain regions during infant development, and can be used as quantified standard reference for detecting rCBF alterations (e.g., the z scores) of atypically developing brains. Elucidating the ontogeny of infant brain physiology and its functional correlates could greatly advance current understanding of general principles of early brain development.

Gradient of functional network maturations in early brain development has been more extensively characterized with recent rs-fMRI studies. Differential emergence of these functional networks is distinguished by different onset time as well as different maturational rate of various brain functions in a given developmental period. For example, primary sensory and motor functional networks, such as the SM and Vis networks, appear earlier before or around birth (*Cao et al., 2017a*; *Doria et al., 2010*;

*Fransson et al., 2007*; *Smyser et al., 2010*; *Peng et al., 2020*). Other functional networks involved in heteromodal functions appear later. The DMN (*Fox et al., 2007*; *Greicius et al., 2003*; *Greicius et al., 2009*; *Raichle et al., 2001*; *Raichle, 2015*; *Smith et al., 2009*) is a higher-order functional network. *Smyser et al., 2010* found that SM and Vis functional networks mature earlier and demonstrate adult-like pattern for preterm neonate brain, with the DMN much immature and incomplete around birth.*Cao et al., 2017a* also found rapid maturation of primary sensorimotor functional systems in preterm neonates from 31 to 41 postmenstrual weeks while the DMN remained immature during that period. These previous studies suggest that significant functional maturation in primary sensorimotor networks occur earlier in preterm and perinatal developmental period (*Cao et al., 2017a*; *Doria et al., 2010*; *Smyser et al., 2010*) compared to 0–24-month infancy focused in this study. Functional network emergence in the DMN was found in the developmental infant cohort in *Figure 1*, marking significant maturation of the DMN in infancy and distinguished network pattern from earlier developmental period. The delineated DMN emergence in this study is also consistent to the literature (*Gao et al., 2009*). *Figure 1—figure supplement 2* further demonstrated significant increase of FC only in the DMN, but not in primary sensorimotor system that already emerged in earlier developmental period.

Glucose and oxygen are two primary molecules for energy metabolism in the brain (*Raichle et al., 2001*; *Vaishnavi et al., 2010*). Glucose consumed by infant brain represents 30% total amount of glucose (*Raichle, 2010*; *Settergren et al., 1976*), more than 15–20% typically seen in adult brain (*Bouma and Muizelaar, 1990*; *Satterthwaite et al., 2014*). The cerebral metabolic rates for glucose (CMRGlu) and oxygen (CMRO2) are direct measures of the rate of energy consumption, which parallel the proliferation of synapses in brain during infancy (*Raichle, 2010*). RCBF delivering glucose and oxygen for energy metabolism in the brain is closely related to CMRGlu and CMRO2 and can serve as a surrogate of these two measurements (*Fox and Raichle, 1986*; *Gur et al., 2009*; *Paulson et al., 2010*; *Vaishnavi et al., 2010*). In the PET study (*Chugani and Phelps, 1986*) using CMRGlu measurements, it was found that the local CMRGlu in the sensorimotor cortex almost reaches the highest level in early infancy and then plateaus during rest of infancy, consistent with relatively small changes of rCBF in later infancy in primary sensorimotor ROIs found in this study (*Figure 3*). On the other hand, the global CBF measured with PC MRI (*Liu et al., 2019*) increases dramatically during infancy with global CBF at 18 months almost five times of the CBF around birth. Taken together, the literature suggests significant but nonuniformly distributed CBF increases across the brain regions during infancy, consistent to the measured heterogeneous rCBF increase pattern (*Figures 2 and 3*) in this study.

Furthermore, the differentiated cerebral metabolic pattern reflected by measured rCBF distribution (*Figure 2*) at a specific age and differential increase rates of the rCBF (*Figure 3*) from 0 to 24 months are strikingly consistent with spatiotemporally differentiated functional (*Cao et al., 2017a*; *Doria et al., 2010*; *Fransson et al., 2007*; *Gao et al., 2009*; *Smyser et al., 2010*) and structural (*Ouyang et al., 2019a*) maturational processes. In developmental brains, cellular processes supporting differential functional emergence require extra oxygen and glucose delivery through cerebral blood flow to meet the metabolic demand. In Hebb's principle, 'neurons that fire together wire together.' Through the synaptogenesis in neuronal maturation, the neurons within a certain functional network system tend to have more synchronized activity in a more mature stage than in an immature stage. As shown in the diagram in *Figure 5—figure supplement 3*, cellular activities in developmental brains, such as synaptogenesis critical for brain circuit formation, need extra energy more than that in the stable and matured stage. Neurons do not have internal reserves of energy in the form of sugar or oxygen. The demand of extra energy requires an increase in rCBF to deliver more oxygen and glucose for the formation of brain networks. In the context of infant brain development, there is a cascade of events of CBF increase, CMRO2 and CMRGlu increase, synaptogenesis and synaptic efficacy increase, blood oxygenation level-dependent (BOLD) signal synchronization increase, and functional connectivity increase, shown in the bottom of *Figure 5—figure supplement 3*. Such spatial correlation of rCBF to the FC in the functional network ROIs was found in adults (*Liang et al., 2013*). Higher rCBF has also been found in the DMN in children 6–20 years of age (*Liu et al., 2018*). Consistent with the diagram shown in *Figure 5—figure supplement 3*, *Figure 4* revealed significant correlation between FC and rCBF in the DMN network, and *Figure 5* identified this significant correlation between FC and rCBF specifically in the DMN network, but not in the primary Vis or SM networks. Collectively, these results

(*Figures 4 and 5*) were well aligned with the hypothesis that faster rCBF increase in the DMN underlies the emergence of the DMN reflected by significant FC increases. The revealed physiology–function relationship may shed light on physiological underpinnings of brain functional network emergence.

It is noteworthy to highlight a few technical details below. First, this study benefits from multi-modal MRI allowing measuring functional network emergence and rCBF of the same cohort of infants. Simultaneous rCBF and FC measurements enabled us to probe relationship of brain physiology and function during infant development. Second, a nonparametric permutation analysis without a prior hypothesis at the voxel level across the whole brain was conducted to confirm the coupling of rCBF and FC is specific in the DMN regions (*Figure 5*), not in the primary sensorimotor (Vis or SM) regions (*Figure 5—figure supplement 2*), demonstrating the robustness of the results on physiological underpinning of functional network emergence in the DMN. Third, the pCASL perfusion MRI was calibrated by PC MRI so that the errors caused by varying labeling efficiency and varying T1 value of arterial blood among infant subjects can be ameliorated. This subject-specific calibration process therefore enhanced the accuracy of rCBF measurements for their potential use as a standard reference. For studies without additional PC MRI acquired for subject-specific calibration, utilizing developmentally appropriate or even subject-specific T1 values of arterial blood is encouraged to improve the accuracy of rCBF measurements. Given the lack of consensus on pCASL acquisition parameters for infants, future research focusing on systematically optimizing the pCASL acquisition protocol in the infant population is needed. Fourth, the ROIs for rCBF measurements were obtained by data-driven ICA of the same sub-cohort of infant subjects aged 12–24 months instead of transferred ROIs from certain brain parcellation atlases. Since most of the parcellation atlases were built based on adult brain data and all these atlases were established based on other subject groups, ROIs delineated from the same cohort improve accuracy of coupling analysis. There are several limitations in this study that can be improved in future investigations. All data were acquired from a cross-sectional cohort. To minimize the inter-subject variability, future study with a longitudinal cohort of infants is warranted. With relatively small size of infant brains, spatial resolution of the pCASL and rs-fMRI can be further improved too to improve imaging measurement accuracy. With relatively larger voxel size of the pCASL scan, partial volume effects could lead to bias of the rCBF measurement. Due to smaller brain and thinner brain cortex of younger infants, using consistent pCASL acquisition voxel size could lead to heterogeneous partial volume effects across infants with different brain sizes and cortical thicknesses. The rCBF biases related to partial volume effects were minimized with calibration of individual PC MRI from all infants. Relatively larger pCASL acquisition voxel size might also contribute to blurring the smaller red areas in the occipital lobe of the brain into a larger continuous red region, although higher rCBF values at the base of the brain and occipital regions than in the frontal regions were observed in infant and neonate brains (e.g., *Kim et al., 2018*; *Lemaître et al., 2021*; *Wang et al., 2008*; *Satterthwaite et al., 2014*). We believe that the rCBF increase pattern during maturation is more complicated than relatively simplified posterior-to-anterior and inferior-to-superior gradients. It also presents a primary-to-association gradient reproducibly found in the literature (see *Sydnor et al., 2021* for review). Such mixed patterns are consistent with the maturation pattern demonstrated in *Figure 3a*. More clear maturation gradient could benefit from future rCBF maps of higher signal-to-noise ratio and higher resolution. To further improve the statistical power, larger infant sample size will be beneficial in the future studies. Finally, although the rCBF from pCASL perfusion MRI is highly correlated with the CMRO2 and CMRGlu measured from PET, rCBF is not a direct measurement of the rate of energy consumption. Physiology–function relationship studies in infants could benefit from the development of novel noninvasive MR imaging methods as an alternative to PET to measure CMRO2 or CMRGlu without the involvement of radioactive tracer.

## Conclusion

Novel findings in this study inform a physiological mechanism of DMN emergence during infancy with rCBF and FC measurements from multimodal MRI in developmental infant brains. The age-specific whole-brain rCBF maps and spatiotemporal rCBF maturational charts in all brain regions serve as a standardized reference of infant brain physiology for precision medicine. The rCBF-FC coupling results revealing fundamental physiology–function relationship have important implications in altered network maturation in developmental brain disorders.

## Materials and methods

### Infant subjects

Forty-eight infants (30 males) aged 0–24 months (14.6 ± 6.32 months) were recruited at Beijing Children's Hospital. These infants were referred to MRI due to simple febrile convulsions (n = 38), diarrhea (n = 9), or sexual precocity (n = 1). All infants had normal neurological examinations documented in medical record. The exclusion criteria include known nervous system disease or history of neurodevelopmental or systemic illness. Every infant's parents provided signed consent, and the protocol was approved by the Beijing Children's Hospital Research Ethics Committee (approval number 2016-36).

### Data acquisition

All infant MR scans, including rs-fMRI, pCASL, PC MRI, and structural MRI, were acquired with the same 3T Philips Achieva system under sedation by orally taken chloral hydrate with dose of 0.5 ml/kg and no more than 10 ml in total. Previous studies (*Li et al., 2011*; *Suzuki et al., 2021*) suggested no significant impact of chloral hydrae on CBF or sensory function. Earplug and headphones were used to minimize noise exposure. Resting state fMRI (rs-fMRI) images were acquired using echo planar imaging with the following parameters: TR = 2000 ms, TE = 24 ms close to TE used for adults since the relative BOLD parameter of infants aged 9 months was found quite close to that of adults (*Cusack et al., 2018*), flip angle = 60°, 37 slices, FOV = 220 × 220 mm$^2$, matrix size = 64 × 64, voxel size = 3.44 × 3.44 × 4 mm$^3$. 200 dynamics were acquired for each infant. The acquisition time of the rs-fMRI images was 7 min. Pseudo-continuous arterial spin labeling (pCASL) perfusion MRI images were acquired using a multi-slice echo planar imaging with the following parameters: TR = 4100 ms, TE = 15 ms, 20 slices with 5 mm slice thickness and no gap between slices, field of view (FOV) = 230 × 230 mm$^2$, matrix size = 84 × 84, voxel size = 2.74 × 2.74 × 5 mm$^3$. As shown on the left panel of *Figure 2a*, the labeling slab was placed at the junction of spinal cord and medulla (65 mm below central slab of imaging volume) and parallel to the anterior commissure-posterior commissure (AC-PC) line. The labeling duration was 1650 ms, and the post labeling delay (PLD) was 1600 ms. Thirty pairs of control and label volume were acquired for each infant. The acquisition time of the pCASL perfusion MRI images was 4.2 min. An auxiliary scan with identical readout module to pCASL but without labeling was acquired for estimating the value of equilibrium magnetization of brain tissue. For accurate and reliable voxel-wise comparison of rCBF across different infant ages, consistent imaging parameters were applied in this study. Imaging parameters, including PLD of infants, were selected closer to those of children and in the interval between imaging parameters of neonates and those of children, with PLD 2000 ms for neonates and 1500 ms for children recommended by the white paper (*Alsop et al., 2015*). The mean arterial transit time (ATT) in infants was reported to be less than 1500 ms (*Varela et al., 2015*; *Kim et al., 2018*). Also following the ASL white paper (*Alsop et al., 2015*) suggestion that the selected PLD should be larger than the ATT, a tailored and consistent PLD of 1600 ms larger than ATT of infants was selected for the studied infant cohort to minimize the effect of PLD on rCBF differences across subjects or across brain regions. PC MRI was acquired to calibrate rCBF by scaling the overall CBF of entire brain. The carotid and vertebral arteries were localized based on a scan of time-of-flight (TOF) angiography acquired with the following parameters: TR = 20 ms, TE = 3.45 ms, flip angle = 30°, 30 slices, FOV = 100 × 100 mm$^2$, matrix size = 100 × 100, voxel size = 1 × 1 × 1 mm$^3$. The acquisition time of the TOF angiography was 28 s. Based on the angiography, the slices for the PC MRI of internal carotid arteries were placed at the level of the foramen magnum and the slices for the PC MRI of vertebral arteries were placed between the two turns in V3 segments, as illustrated on the right panel of *Figure 2a*. For left and right internal carotid and vertebral arteries, PC MRI images were acquired with the following parameters: TR = 20 ms, TE = 10.6 ms, flip angle = 15°, single slice, FOV = 120 × 120 mm$^2$, matrix size = 200 × 200, voxel size = 0.6 × 0.6 × 3 mm, maximum velocity encoding = 40 cm/s, non-gated, four repetitions. The acquisition time of PC MRI for each artery was 24 s. T1-weighted images of all infants were also acquired for anatomical information and brain segmentation using MPRAGE (Magnetization Prepared – RApid Gradient Echo) sequence with the following parameters: TR = 8.28 ms, TE = 3.82 ms, flip angle = 12°, TI (time of inversion) = 1100 ms, 150 slices, FOV = 200 × 200 mm$^2$, matrix size = 200 × 200, voxel size = 1 × 1 × 1 mm$^3$. The acquisition time of the T1-weighted image was 3.7 min. Visual inspection was carefully conducted for all MRI data by experienced pediatric radiologists (DH and YP) with decades of experience in clinical radiology. No

significant motion artifacts were spotted with the sedated MRI scans. Therefore, no dataset from the 48 infants was excluded from the following data analysis due to severe motion artifacts.

## Rs-fMRI preprocessing

The same preprocessing procedures elaborated in our previous publication (*Cao et al., 2017b*) was used. Briefly, the normalized rs-fMRI images underwent spatial smoothing with a Gaussian kernel of full width at half-maximum (FWHM) of 4 mm, linear trend removal, and temporal band-pass filtering (0.01–0.10 Hz). Several nuisance variables, including six rigid-body head motion parameters and the averaged signal from white matter and cerebrospinal fluid (CSF) tissue, were removed through multiple linear regression analysis to reduce the effects of non-neuronal signals. The CSF and white matter were segmented with the T1-weighted image using SPM. Preprocessed rs-fMRI signals were used to estimate functional connectivity (FC), defined as Pearson's correlation between the time courses of preprocessed (rs-fMRI BOLD) signal in two regions or two voxels (*Figure 1—figure supplement 1a*).

## Identification of functional network ROIs with rs-fMRI

Brain regions of three functional networks, including DMN, Vis network, and SM network, were used as functional ROIs for quantifying rCBF. These functional networks were identified with rs-fMRI of 35 infants aged 12–24 months as DMN can be better delineated in later infancy than earlier infancy. ICA (*Beckmann et al., 2005*) in the FSL (https://fsl.fmrib.ox.ac.uk/fsl/fslwiki/MELODIC) was used to identify all these network regions. Individual DMN regions, including PCC, MPFC, IPL, and LTC, were extracted from the independent components (ICs) based on the spatially distributed regions consistently identified across studies (*Greicius et al., 2004*; *Smith et al., 2009*). Vis and SM network regions were also identified following the literature (*Smith et al., 2009*). The functional network ROIs were obtained by thresholding the ICs at z value of 1.96, which corresponds to p value of 0.05. The identified ROIs for DMN, Vis, and SM networks are shown in *Figure 1—figure supplement 1b*. ROIs of DMN, Vis, or SM include all regions in each network, respectively. The functional network ROIs overlapped with the binary cortical mask in the template space were used for measuring ROI-based rCBF below.

## Measurement of rCBF with pCASL perfusion MRI and calibrated by PC MRI

After head motion correction of the pCASL perfusion MRI, we estimated rCBF using the protocol similar to that in our previous publication (*Ouyang et al., 2017*). Briefly, rCBF was measured using a model described in ASL white paper (*Alsop et al., 2015*):

$$rCBF = \frac{6000 \cdot \lambda \cdot \Delta M \cdot e^{\frac{PLD}{T_{1a}}}}{2 \cdot \alpha \cdot M_b^0 \cdot T_{1a} \cdot \left(1 - e^{\frac{-LabelDur}{T_{1a}}}\right)} \; [ml/100g/min] \tag{1}$$

where ΔM is the dynamic-averaged signal intensity difference between in the control and label images; $\lambda$, the blood–brain partition coefficient, is 0.9 ml/g (*Herscovitch and Raichle, 1985*); PLD, the post labeling delay time, is the cumulation of 1650 ms and the delayed time between slices; LabelDur, the labeling duration, is 1600 ms; α, the labeling efficiency, is 0.86 predicted by the fitting between labeling efficiency and blood velocity in the previous study (*Aslan et al., 2010*); $T_{1a}$, $T_1$ of arterial blood, is 1800 ms (*Liu et al., 2016*; *Varela et al., 2011*). The value of equilibrium magnetization of brain tissue ($M_b^0$) was obtained from an auxiliary scan with identical readout module of pCASL except labeling. The labeling efficiency α can vary considerably across participants, especially in infants. Thus, we used PC MRI to estimate and calibrate rCBF measures, as described previously (*Aslan et al., 2010*; *Ouyang et al., 2017*). To calibrate rCBF, global CBF from PC MRI was calculated as follows:

$$f_{PC,AVG} = \int v \, dA/(\rho * brainvolume) \tag{2}$$

where v is the blood flow velocity in the ICAs and VAs; A is the cross-sectional area of the blood vessel with the unit mm²; and the brain tissue density $\rho$ is assumed as 1.06 g/mL (*Dittmer and*

*Dawson, 1961*; *Herscovitch and Raichle, 1985*). Brain volume was measured from the T1-weighted image as parenchyma volume (gray matter + white matter volume). RCBF was calibrated by applying the scalar factor making averaged rCBF equal to global CBF from PC MRI. To demonstrate the reproducibility of the adopted pCASL protocol, the ICC was calculated based on entire brain rCBF maps measured from first half and second half of control/label series of pCASL scan of a randomly selected infant subject aged 17.6 months.

## Multimodal image registration to a customized template space from all subjects

For integrating perfusion MRI and rs-fMRI data of all infant subjects, a customized structural template was generated. T1-weighted image of a 12-month-old brain characterized by median brain size at this age and straight medial longitudinal fissure was used as a single-subject template. T1-weighted images of all subjects were registered to the single-subject template by using nonlinear registration in Statistical Parametric Mapping (SPM 8, http://www.fil.ion.ucl.ac.uk/spm). The averaged T1-weighted image in the template space was defined as the structural template. In individual space, segmentation of brain gray matter was also conducted using the contrasts of T1-weighted image with SPM, generating the gray matter tissue probability map in individual space. After slice timing and head motion correction, intra-subject registration of rs-fMRI to T1-weighted image in the individual space was conducted by transforming the averaged images across dynamics in each session to the T1-weighted image of the same subject through linear registration with SPM. This intra-subject transformation was applied to each volume of rs-fMRI. As described in the main text, rCBF map was estimated in the individual space using Eq (1). Intra-subject registration of rCBF map to T1-weighted image was also conducted through linear registration with SPM. RCBF maps, gray matter probability map, and rs-fMRI images aligned to T1-weighted images in the individual space of each subject were then normalized to the customized infant template space through the same nonlinear registration from individual T1-weighted image to the structural template. After nonlinear inter-subject normalization, rCBF map was smoothed spatially with Gaussian kernel of FWHM 4 mm in the customized template space. An averaged gray matter probability map was also generated after inter-subject normalization. After carefully testing multiple thresholds in the averaged gray matter probability maps, 40% probability minimizing the contamination of white matter and CSF while keeping the continuity of the cortical gray matter mask was used to generate the binary cortical mask.

## Characterization of age-dependent changes of rCBF and FC

After multimodal images from all subjects were registered in the same template space, age-dependent rCBF and FC changes were characterized using linear regression. The linear cross-sectional developmental trajectory of voxel-wise rCBF increase was obtained by fitting rCBF measurements in each voxel with ages across subjects. The fitted rCBF at different ages was estimated and projected onto the template cortical surface with Amira (FEI, Hillsboro, OR) to show the spatiotemporal changes of rCBF across cortical regions during infancy. The rCBF–age correlation coefficients r values of all voxels were estimated and mapped to the cortical surface, resulting in the correlation coefficient map. The functional ROIs identified above were used to quantify regional rCBF change. The rCBF in the SM, Vis, and DMN ROIs were averaged across voxels within each ROI, and fitted with a linear model: $rCBF(t) = \alpha + \beta t + \varepsilon$, where $\alpha$ and $\beta$ are intercepts and slopes for rCBF measured at certain ROI, t is the infant age in months, and $\varepsilon$ is the error term. To test whether regional rCBF increased significantly with age, the null hypothesis was that rCBF slope in each ROI was equal to zero. To compare regional rCBF change rates between the tested ROI and SM ROI, the null hypothesis was that the rCBF slope of the tested ROI and rCBF slope of SM ROI were equal. Rejection of the null hypothesis indicated a significant rCBF slope difference between two ROIs.

Nonlinear models, including exponential and biphasic models, were compared with linear model using F-test for fitting regional rCBF in the DMN, SM, and Vis ROIs. No significant difference was found between linear and exponential (all $F(1,45) < 3.45$, $p>0.05$, uncorrected) fitting or between linear and biphasic (all $F(2,44) < 2.64$, $p>0.05$, uncorrected) fitting in the regional rCBF changes in the DMN, SM, and Vis ROIs.

In the customized template space, time-dependent changes of FC to PCC and time-dependent changes of FC within the functional network ROIs were calculated. FC of any given voxel outside the

PCC to PCC was defined as correlation between signal time course of this given voxel and the averaged signal time course of all voxels within the PCC. Emergence of the DMN was delineated by linear fitting of the age-related increase of FC to PCC in each voxel across subjects. FC within each network was defined as the mean of functional connectivity strength (*Cao et al., 2017b*) of each voxel within this network region. The FC within the DMN, SM, or Vis was also fitted with a linear model: $FC(t) = \alpha + \beta t + \varepsilon$, where $\alpha$ and $\beta$ are intercepts and slopes for FC measured within a certain network, t is the infant age in months, and $\varepsilon$ is the error term. To test whether the FC within a network increased significantly with age, the null hypothesis was that rCBF slope in each ROI was equal to zero.

### Test of heterogeneity of rCBF across functional network ROIs

To examine heterogeneity of infant rCBF in different brain regions, rCBF measurements of all infants were averaged across voxels in the Vis, SM, and DMN (including DMN subregions DMN_PCC, DMN_MPFC, DMN_IPL, and DMN_LTC), respectively. To test significant difference of the rCBF values among different functional network ROIs, a one-way ANOVA with repeated measures was conducted. Paired *t*-tests were also conducted to test the difference of rCBF measurements between regions. FDR of each test was corrected to control the type I error. Significant interaction between regions and age was tested with an ANCOVA test where age was used as a covariate.

### Coupling between rCBF and FC during the infant brain development

Coupling between rCBF and FC in the DMN was conducted with voxel-wise approach. The FC of a voxel in the DMN was the average of correlations of rs-fMRI BOLD signal between this voxel and all other DMN voxels. All infants were divided into two groups based on their ages, 0–12 months and 12–24 months. 4000 voxels were randomly chosen from the DMN voxels of all subjects in each age group for correlation analysis. Since the variance of both FC and rCBF cannot be ignored in this study, Deming regression (*Deming, 1943*) was used to fit the trendline of coupling between FC and rCBF. Partial correlation between rCBF and FC regressing out age effects was also conducted to confirm the significant rCBF-FC coupling in the DMN excluding the age effects. We further tested whether significant FC-rCBF coupling was specifically localized in the DMN, but not in primary sensorimotor (Vis or SM) regions. FC within a specific network was calculated by averaged FC of all voxels in this network. As demonstrated in *Figure 5—figure supplement 1*, the correlation between FC within a network and the rCBF at each voxel resulted in a whole-brain r map (e.g., *Figure 5b*). A nonparametric permutation test was then applied to evaluate the significance of rCBF–FC correlation in the ROIs of a specific brain network (e.g., DMN, SM, or Vis). The null hypothesis is that the voxels with significant correlation ($r > 0.28$) between rCBF and FC are distributed evenly in the brain. To test the null hypothesis, we resampled the correlation coefficient r of all brain voxels randomly for 10,000 times to build 10,000 whole-brain correlation coefficient distribution maps. The FC–rCBF correlation was considered significant in certain brain network ROIs if the number of observed significant voxels in the network ROIs is higher than the number of significant voxels corresponding to 95th percentile in the permutation tests.

## Acknowledgements

This work was supported by grants from the National Institute of Health: R01MH092535, R01MH125333, R01EB031284, R21MH123930, and P50HD105354.

## Additional information

### Funding

| Funder | Grant reference number | Author |
| --- | --- | --- |
| National Institutes of Health | R01EB031284 | Hao Huang |
| National Institutes of Health | R21MH123930 | Minhui Ouyang |

| Funder | Grant reference number | Author |
|---|---|---|
| National Institutes of Health | R01MH125333 | Hao Huang |
| National Institutes of Health | R01MH092535 | Hao Huang |
| National Institutes of Health | P50HD105354 | Hao Huang |
| National Institutes of Health | R01MH129981 | Hao Huang |

The funders had no role in study design, data collection and interpretation, or the decision to submit the work for publication.

## Author contributions

Qinlin Yu, Software, Formal analysis, Validation, Investigation, Visualization, Methodology, Writing - original draft, Writing - review and editing; Minhui Ouyang, Software, Formal analysis, Validation, Visualization, Methodology, Writing - original draft, Writing - review and editing; John Detre, Bo Hong, Fang Fang, Methodology, Writing - review and editing; Huiying Kang, Di Hu, Yun Peng, Resources, Data curation, Project administration, Writing - review and editing; Hao Huang, Conceptualization, Resources, Data curation, Software, Formal analysis, Supervision, Funding acquisition, Validation, Investigation, Visualization, Methodology, Writing - original draft, Project administration, Writing - review and editing

## Author ORCIDs

Minhui Ouyang http://orcid.org/0000-0001-8013-2553
Bo Hong http://orcid.org/0000-0003-2900-6791
Hao Huang http://orcid.org/0000-0002-9103-4382

## Ethics

Informed parental consents were obtained from the subject's parent. The Institutional Review Board of Beijing Children's Hospital Research Ethics Committee (Approval number 2016-36) approved the study procedures.

## Decision letter and Author response

Decision letter https://doi.org/10.7554/eLife.78397.sa1
Author response https://doi.org/10.7554/eLife.78397.sa2

# Additional files

## Supplementary files

- MDAR checklist

## Data availability

Maps including regional cerebral blood flow (rCBF) maps and functional connectivity maps from multimodal infant MRI datasets including pseudo-continuous arterial-spin-labelled perfusion MRI and resting-state MRI of forty-eight infants are publicly available from Huang lab GitHub repository (https://github.com/haohuanglab/infant_perfusion_function, copy archived at swh:1:rev:34071e49232d7960ff6b78464b55077da136d868). However, we cannot openly share the raw, unprocessed MRI data, because The Institutional Review Board of Beijing Children's Hospital Research Ethics Committee (Approval number 2016-36) specifies the participants did not give consent for these data to be released publicly. The raw MRI data can be made available to individual researchers on informal request to the corresponding author through email hao.huang@pennmedicine.upenn.edu. Source code used in analysis and related documentation are also available at the Huang lab GitHub repository (https://github.com/haohuanglab/infant_perfusion_function).

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
