## [Editor Report]

In this paper, the authors find a link between the emergence of functional connectivity (FC) and changes in regional Cerebral Blood Flow (rCBF) in human infancy from birth to 24 months of age, which will be of interest to the increasing field investigating how the establishment of the brain's functional organization is linked to neurodevelopmental and psychiatric conditions. The data quality and complementarity are impressive for infants over this developmental period (0-2 years). Most of the key claims of the manuscript are well supported by the data. However, the relatively sparse sample and cross-sectional nature do limit interpretation.

---

## [Decision Letter]

**Decision letter after peer review:**

Thank you for submitting your article "Infant brain regional cerebral blood flow dynamics supporting emergence of vital functional networks" for consideration by *eLife*. Your article has been reviewed by 3 peer reviewers, and the evaluation has been overseen by a Reviewing Editor and Timothy Behrens as the Senior Editor. The following individual involved in review of your submission has agreed to reveal their identity: Jessica Dubois (Reviewer #1).

Essential revisions:

The reviewers found much to like about this study, but also raised some concerns. I am including the individual reviews below, but I am going to highlight the ones that will be a particular concern to the reviewers on revision.

1) There is a general concern that the rCBF differences you see across brain regions (values that seem to follow a spatial gradient from inferior to superior and from posterior to anterior) might reflect a technical issue/bias in relation to anatomy and physiological changes, as opposed to specific maturational changes and differences across brain networks including the DMN. This is alluded to in several of the reviewer comments, particularly but not exclusively in R3. Several strategies are suggested for alleviating this concern.

2) The use of the same imaging parameters across the population might lead to an underestimation of rCBF in the younger infants. If there is evidence against this, it should be supplied.

3) Similarly using the older infants to get the ROIs may lead to a selection bias. R2 has suggestions for unbiased analysis strategies.

4) The manuscript sells itself as a description of connectivity, broadly defined, but focuses on DMN. All reviewers think this needs clear justification in the intro, and potentially changes to the title and abstract. The word "vital" should also be removed from the title.

*Reviewer #1 (Recommendations for the authors):*

– Regarding rCBF measures in DMN network, it would be interesting to evaluate correlations across DMN regions: this might highlight that the joint functioning of these regions requires similar changes in rCBF across ages, which would mean a kind of "physiological connectivity" throughout development.

– Evaluating possible rCBF asymmetries between left and right hemispheres would be interesting given previous observations of the two following studies (currently missing in the list of references):

Chiron C, Jambaque I, Nabbout R, Lounes R, Syrota A, Dulac O. The right brain hemisphere is dominant in human infants. Brain. 1997 Jun;120 ( Pt 6):1057-65. doi: 10.1093/brain/120.6.1057.

Lemaître H, Augé P, Saitovitch A, Vinçon-Leite A, Tacchella JM, Fillon L, Calmon R, Dangouloff-Ros V, Lévy R, Grévent D, Brunelle F, Boddaert N, Zilbovicius M. Rest Functional Brain Maturation during the First Year of Life. Cereb Cortex. 2021 Feb 5;31(3):1776-1785. doi: 10.1093/cercor/bhaa325.

*Reviewer #2 (Recommendations for the authors):*

1. Whilst I understand the focus of the study on putative functional networks which have been shown to mature at different rates, this approach does limit the study somewhat. The title and the first part of the introduction suggest that the work is about whole brain development, but then the abstract and paper suddenly focus on the default mode network. Whilst the reason for this focus becomes apparent with reading, I found the motivation for their approach therefore somewhat unclear/inconsistent.

2. Could the authors better describe in the mansucript why they chose to focus only on three networks rather than whole brain approaches or suggest what they might consider doing (such as linked ICA together with anatomical data) in future studies given the available data?

3. With the networks specifically, it is possible that the use of the DMN maps derived from the older infants only, may bias the results as it could then be argued that it is unsurprising that connectivity (and potentially rCBF is less) in those infants. Would it be possible to see if an age-specific representation of the network (using a method such as dual regression) would still yield the same results?

4. It is mentioned in the results that the infants were sedated for study – the type of sedation should be mentioned (the drug and dose) and the possible implications on the results (especially the CBF) considered. This is also the case for the clinical history of the infants – a large proportion of them had a history of seizures, which is a concern as it questions the generalisability of the results to the healthy brain (even if they are apparently neurologically normal).

5. As mentioned above, it is a concern that the same acquisition and processing parameters were used for all of the infants. The T1 of arterial blood for example, seems to have been derived from published neonatal values which are likely to differ greatly from those needed for a 2 year old infant. In contrast, arterial transit time is likely to vary across development, and so the use of the same post-label delay time of 1600ms (which is markedly shorter than the value of 2000ms suggested in the Alsop white paper) for all infants is a concern – as this may lead to an underestimation of CBF in the youngest infants. Partial voluming effects will be worse in the youngest infants (especially with a slice thickness of 5mm), which could also lead to an underestimation of CBF as separation of the white matter and cortex is challenging.

6. I find the suggestion that the rCBF measures can tell us about changes in brain metabolism rather speculative – as there are no other measures of oxygen extraction for example to specifically justify this. As the data is cross-sectional, mechanistic insight such as understanding whether one precedes the other is not possible. Figure 6 for example is not justified by the data, as it is simply not possible to know from their data if the suggested physiological changes are actually taking place.

7. The language in parts is a little unclear and therefore could benefit from clarification. Example include: line 53 "increasing feed of energy to fuel the brain development" or line 56 "impeding fundamental view of energy expenditure across functional systems of early developing brain"

Specific points:

1. I would prefer rewording of the title if possible – the use of the word "vital" does not seem appropriate, as this is not demonstrated in the work.

2. In the first line of the abstract, the sentence reads rapid cerebral blood flow (rCBF) which incorrectly implies that this is what the acronym stands for.

3. The abstract mentions "rCBF dynamics" – which I would caution against. In this work, they are really only looking at the mean rCBF across the acquisition, rather than its specific dynamics across the volumes acquired.

4. Suggest rewording of line 30 "We found faster rCBF increases in the DMN than other regions." as whilst this is true in comparison to the visual/SM networks – it wasn't demonstrated in comparison to all other regions?

5. The statement in line 122 of "high quality of the rCBF maps" is not justified as its not clear how this "quality" has been defined – suggest rephrasing, please.

6. The values of rCBF on line 141 should state they are the mean across all of the infants and should include some estimate of the variance.

7. I note that rCBF is consistently higher at the base of the brain and occipital regions – is this a recognized pattern in the literature? If not, could it be because vascular crushers were not used and there is overestimation due to the residual blood in the basilar artery?

*Reviewer #3 (Recommendations for the authors):*

The manuscript describes a study of 48 infants aged 0-24 months who underwent resting state functional MRI (rs-fcMRI) and arterial spin labeling regional blood flow (rCBF) measurements. The authors focused on the default mode network (DMN), visual cortex (VIS), and somatomotor cortex (SM). They found faster increases in rCBF in the DMN with strongly coupled increases of rCBF to network "strength" measured through rs-fcMRI. They concluded that the faster increase in rCBF in the DMN takes place to meet the metabolic demands of DMN maturation.

The authors found stronger correlations between increases in rCBF and rs-fcMRI in the DMN than in VIS and SM. They suggest that this indicates that changes in rCBF constitute a physiological underpinning of DMN maturation. However, it is not clear that the underlying assertion that increases in rs-fcMRI (developmental maturation) and increases in rCBF are tightly linked is supported by the data.

First, VIS and SM are both more mature than DMN at birth, yet VIS has the highest rCBF and SM has the lowest. If rCBF and maturity were tightly coupled, one would expect both to have relatively high rCBF values.

Second, the data shown in Figure 2A show that while rCBF increases with maturation, it does so along posterior-to-anterior and inferior-to-superior gradients which do not appear to correlate with regional variability in maturation. The SM region chosen for the study is located mainly in the leg area, which is primarily in the superior and medial SM cortex. The more inferior SM areas, corresponding to the arm and face, are not included. Given an inferior-to-superior gradient in the maturational increase in rCBF, the more superior portion of SM chosen is relatively late to show an increase in rCBF.

Third, there are fairly large differences in the rCBF of subregions of the DMN, though one would expect them to be similar if they are maturing as a unit.

Finally, the stronger correlation between increases in rCBF and rs-fcMRI metrics for the DMN may reflect its greater increase in rCBF during the time period studied. This may be due to the finding that VIS, located posteriorly and inferiorly, has already undergone its greatest increase in rCBF; while SM, located superiorly, has yet to undergo its greatest increase in rCBF. Thus both have relatively modest changes in rCBF during the period studied. The steeper rise in rCBF in the DMN, which may be by virtue of its location, would allow for a stronger correlation with maturational increases in rs-fcMRI parameters.

Since the gradient in increasing rCBF during development appears to be based on relative location, it is important to consider whether the superior-inferior gradient might be a technical artifact related to applying labeling pulses to the neck, which is getting longer, or changes in flow velocities with age.

[Editors’ note: further revisions were suggested prior to acceptance, as described below.]

Thank you for resubmitting your work entitled "Infant brain regional cerebral blood flow increases supporting emergence of the default-mode network" for further consideration by *eLife*. Your revised article has been evaluated by Timothy Behrens (Senior Editor) and a Reviewing Editor.

The manuscript has been improved but there are some remaining issues that need to be addressed, as outlined below:

As you will see the reviewers have some concerns remaining. The largest concern is that measurement differences in acquisition might have introduced bias. It would clearly be best if you could provide some evidence, or even simulation, to address these points (detailed below), but if this is not possible then please explicitly acknowledge these concerns in the discussion and discuss the possible effects of this bias on your results.

*Reviewer #1 (Recommendations for the authors):*

The authors have partly responded to the reviewers' comments in my opinion. Some points still seem to me to be questionable in the manuscript, whereas some answers provided by the authors in their replies would have benefited from being added to the manuscript.

– Essential revisions 1 and 2: unless I am mistaken, these elements have not been added to the manuscript. Perhaps this would be relevant for readers who don't read the reviewers' comments and the authors' answers in addition to the article.

– R1-1: The answer provided might seem insufficient. The grey matter masks shown seem to me to clearly contain white matter, at least in the youngest infants for whom the cortex is very thin. The problem of reliably identifying the cortex to limit partial volume effects is definitely complex, and pCASL acquisition with thick slices doesn't avoid this problem anyway. That said, I still trust the measures performed, but I think it is important to discuss this point so that the reader is aware of this potential measurement bias. One way of doing this would be to estimate what the bias might be as a function of age (based on the estimated proportions of grey/white matter in the masks considered, and estimated rCBF in each tissue).

– R1-2: Again, the answer provided seems incomplete. When two measures depend on the same variable (age), their correlation can be partly driven by the common correlation, and it cannot then be considered as specific. The similarity between figures 3b and 5d on the lateral side draws attention to this point. Unless I am mistaken, the measures of functional connectivity in the visual and sensorimotor networks are much less age-dependent than in the DMN network (Figure 1-sup2), whereas the rCBF is age-dependent in all regions: this difference might drive the difference in FC-rCBF correlation observed across networks. A simple way to "prove" that the FC-rCBF correlation is indeed specific for the DMN would be to perform all these correlation analyses after removing all age effects in all measurements (region-wise or vertex-wise). These results would be convincing.

– R1-4: it seems to me that the order of the method paragraphs should be changed to follow the new order of the results.

– R1-11: citation of the Cusack et al. paper would be welcome in the manuscript.

– Introduction p3: the sentence "there have been no known whole-brain mappings of heterogeneous infant brain regional cerebral blood flow (rCBF) changes across landmark infant ages thus far" is incorrect (see the Lemaitre et al. paper pointed out in the previous review).

*Reviewer #2 (Recommendations for the authors):*

Thank you for the detailed answers to the questions and amendments to the manuscript. Whilst I agree with some of their responses, I feel there are still some unresolved issues which I think are fundamental to resolve:

1. The use of similar acquisition parameters for all age groups: as far I understand it, the method proposed by Aslan et al. using PCA to calculate the labelling efficiency still requires the T1 of arterial blood as a parameter – for which neonatal values (according to their references) have been used. It would be good to know what would happen if more developmentally appropriate values were used in their CBF estimations. Secondly, the Aslan method provides a single corrective factor (and estimate of the labelling efficiency) for the CBF estimation based on the PCA measurement (which is global). Therefore it cannot correct for any regional effects of using a different post-label delay in the ASL acquisition – with knock-on effects for the regional CBF estimation (such as the spatial bias leading to the inferior-superior gradient for example). Would it be possible for the authors to provide some empirical evidence that the post-label delay has not influenced their results?

2. With regards to the inferior-superior gradient, the references that the authors cite do not show the same kind of effects. Whilst many of them do suggest that CBF is higher posteriorly and is higher in the occipital lobe – it is not a "hot spot" in the same way as seen in the maps here (especially at the youngest ages). The maps in the Lamaitre work for example look very different (see figure 2 in their paper) to those shown in figure 3 here.

3. I am unconvinced by the argument that calculating grey matter maps in native high-resolution space and registering them together with the low-resolution CBF map into standard space thresholded probabilistic maps resolve the partial voluming issue. Fundamentally this does not change the base acquisition voxel which is very large and is the same across all of the infants studied – so differences relative to the size of the brain/cortex and partial voluming in the acquisition itself in the acquisition voxel are surely still an issue?

---

## [Author Response]

Essential revisions:The reviewers found much to like about this study, but also raised some concerns. I am including the individual reviews below, but I am going to highlight the ones that will be a particular concern to the reviewers on revision.1) There is a general concern that the rCBF differences you see across brain regions (values that seem to follow a spatial gradient from inferior to superior and from posterior to anterior) might reflect a technical issue/bias in relation to anatomy and physiological changes, as opposed to specific maturational changes and differences across brain networks including the DMN. This is alluded to in several of the reviewer comments, particularly but not exclusively in R3. Several strategies are suggested for alleviating this concern.

We thank the reviewers for this comment. When initially inspecting the data, we too suspect the spatial gradient might reflect a technical bias in relation to anatomy and physiological changes. We carefully reviewed data before the original submission. We respectively disagree that rCBF difference across brain regions reflects a technical issue/bias due to following justification. The pattern of rCBF spatial gradient from inferior to superior and from posterior to anterior has also been consistently reported in various studies across different age groups. For example, previous studies have observed higher rCBF values at the base of the brain and occipital regions than in the frontal regions in infant and neonate brains (e.g., Kim et al., 2018; Lemaître et al., 2021; Wang et al., 2008). This spatial gradient pattern has also been reported in Satterthwaite’s study with rCBF results of 922 youths aged 8-22 years (Satterthwaite et al., 2014). Consistency across platform and across studies would only indicate bias of arterial spin labelling (ASL) perfusion MRI technique in general. However, ASL perfusion MRI is a noninvasive quantitative method that has been validated extensively against “gold standard” 15O-PET as well as across platforms, including it reproducibility. Although future investigation of rCBF measurements from infants of a similar age range with a larger sample size is warranted, existing literature indicates high rCBF values at the base of the brain and occipital regions are unlikely due to the residual blood in the basilar artery.

Kim, H.G., Lee, J.H., Choi, J.W., Han, M., Gho, S.M., and Moon, Y., (2018). Multidelay arterial spin-labeling MRI in neonates and infants: cerebral perfusion changes during brain maturation. American Journal of Neuroradiology 39(10): 1912-1918.

Lemaître, H., Augé, P., Saitovitch, A., Vinçon-Leite, A., Tacchella, J.M., Fillon, L., Calmon, R., Dangouloff-Ros, V., Lévy, R., Grévent, D., et al. (2021). Rest functional brain maturation during the first year of life. Cereb. Cortex 31, 1776-1785.

Satterthwaite, T.D., Shinohara, R.T., Wolf, D.H., Hopson, R.D., Elliott, M.A., Vandekar, S.N., Ruparel, K., Calkins, M.E., Roalf, D.R., Gennatas, E.D., et al. (2014). Impact of puberty on the evolution of cerebral perfusion during adolescence. Proc. Natl. Acad. Sci. USA 111, 8643-8648.

Wang, Z., Fernández-Seara, M., Alsop, D.C., Liu, W.C., Flax, J.F., Benasich, A.A., and Detre, J.A., (2008). Assessment of functional development in normal infant brain using arterial spin labeled perfusion MRI. Neuroimage 39(3), 973-978.

2) The use of the same imaging parameters across the population might lead to an underestimation of rCBF in the younger infants. If there is evidence against this, it should be supplied.

The reviewer’s comment is valid. We would like to address the comment in the following aspects: (1) infant physiological parameters including PLD and imaging parameter selections are in the interval between neonates and children, and probably closer to children. For example, according to the Alsop white paper, a PLD of 2000ms was recommended for neonates, while a PLD of 1500ms was recommended for children. In our cohort, a PLD time of 1600ms was tailored for the studied infant age range. (2) Extra phase contrast MRI was acquired for all infants. And all rCBF maps of individual infants obtained with ASL scans in our study were calibrated using phase contrast MRI by applying a normalization factor so that there should not be any underestimation of rCBF in the younger infants. As suggested by Aslan et al. (Aslan et al., 2010), this approach largely minimizes the effects of individual variations in processing parameters such as T1 of arterial blood or labeling efficiency in the final estimation of rCBF measurement. Please also see our response to R2-5 for details.

Aslan, S., Xu, F., Wang, P.L., Uh, J., Yezhuvath, U.S., Van Osch, M., and Lu, H. (2010). Estimation of labeling efficiency in pseudocontinuous arterial spin labeling. Magn Reson Med. 63, 765-771.

3) Similarly using the older infants to get the ROIs may lead to a selection bias. R2 has suggestions for unbiased analysis strategies.

We thank the reviewer for this comment. The reviewer raised a very good point. We have strong counter example showing consistent network ROIs generated from older infant data do not lead to biased higher functional connectivity in the older infants, as shown in nonsignificant age-related change of functional connectivity within the visual (Vis) and somatosensory (SM) networks in Figure 1—figure supplement 2.

We had carefully evaluated approach of consistent ROIs used in the manuscript and approach of age-specific ROIs before adopting consistent ROI approach before original submission. Age-specific ROIs may lead to inconsistent measurement and add artificial bias to the measurement aimed for quantifying age-related blood flow changes. Specifically, if different and age-specific ROIs were applied, it would be difficult to delineate if the blood flow change is due to age-related brain maturation or ROI change. Secondly, DMN is not reproducible and reliable at the younger infants characterized by weak functional connections between PCC and other DMN regions (Alcauter et al., 2014). It is not feasible to establish age-specific DMN ROIs for younger infants when the ROIs cannot be reliably identified. And including younger infants to generate ROIs adds extra noise resulting in inaccurate ROIs.

Alcauter, S., Lin, W., Smith, J.K., Short, S.J., Goldman, B.D., Reznick, J.S., Gilmore, J.H., and Gao, W. (2014). Development of thalamocortical connectivity during infancy and its cognitive correlations. Journal of Neuroscience 34(27), 9067-9075.

4) The manuscript sells itself as a description of connectivity, broadly defined, but focuses on DMN. All reviewers think this needs clear justification in the intro, and potentially changes to the title and abstract. The word "vital" should also be removed from the title.

We thank the reviewers for this comment. We used broadly defined network as other network (visual and somatosensory) results were also presented in the manuscript. But we agree those networks were only used as reference networks for focused DMN. Therefore, following suggestions, we have renamed title as “Infant brain regional cerebral flow increases supporting emergence of the default-mode network” and modified abstract and manuscript accordingly too. Please also see our responses to R1-5 and R2-8.

Reviewer #1 (Recommendations for the authors):– Regarding rCBF measures in DMN network, it would be interesting to evaluate correlations across DMN regions: this might highlight that the joint functioning of these regions requires similar changes in rCBF across ages, which would mean a kind of "physiological connectivity" throughout development.– Evaluating possible rCBF asymmetries between left and right hemispheres would be interesting given previous observations of the two following studies (currently missing in the list of references):Chiron C, Jambaque I, Nabbout R, Lounes R, Syrota A, Dulac O. The right brain hemisphere is dominant in human infants. Brain. 1997 Jun;120 ( Pt 6):1057-65. doi: 10.1093/brain/120.6.1057.Lemaître H, Augé P, Saitovitch A, Vinçon-Leite A, Tacchella JM, Fillon L, Calmon R, Dangouloff-Ros V, Lévy R, Grévent D, Brunelle F, Boddaert N, Zilbovicius M. Rest Functional Brain Maturation during the First Year of Life. Cereb Cortex. 2021 Feb 5;31(3):1776-1785. doi: 10.1093/cercor/bhaa325.

We thank the reviewer for this comment. To reveal sort of “physiological connectivity”, the region-wise rCBF correlation was conducted across infants, we found that all the correlations were significant (p<0.001, FDR corrected), with higher correlations between the DMN subregions (i.e. DMN_PCC, DMN_MPFC, DMN_IPL, DMN_ITC), as shown in Author response image 1.

**Author response image 1. sa2fig1:** Physiological connectivity.

To evaluate possible rCBF asymmetry, we compared rCBF in the same brain network regions between two hemispheres, with the results shown in Author response image 3. Significantly higher rCBF (t=3.82, p <0.05) was found in the SM network regions in the right hemisphere (50.8 ± 1.67 ml/100g/min), compared to that in the left hemisphere (47.8 ± 1.43 ml/100g/min). The finding of rCBF asymmetry is consistent with the previous study (Chiron et al., 1997; Lemaître et al., 2021) and has been added to the *Faster rCBF increases in the DMN hub regions during infant brain development* subsection in RESULT section.

Chiron, C., Jambaque, I., Nabbout, R., Lounes, R., Syrota, A., and Dulac, O. (1997). The right brain hemisphere is dominant in human infants. Brain: a journal of neurology, 120, 1057-1065.

Lemaître, H., Augé, P., Saitovitch, A., Vinçon-Leite, A., Tacchella, J.M., Fillon, L., Calmon, R., Dangouloff-Ros, V., Lévy, R., Grévent, D., et al. (2021). Rest functional brain maturation during the first year of life. Cereb. Cortex 31, 1776-1785.

**Author response image 2. sa2fig2:** Significant higher rCBF in the sensorimotor (SM) network regions in right hemisphere.

Reviewer #2 (Recommendations for the authors):1. Whilst I understand the focus of the study on putative functional networks which have been shown to mature at different rates, this approach does limit the study somewhat. The title and the first part of the introduction suggest that the work is about whole brain development, but then the abstract and paper suddenly focus on the default mode network. Whilst the reason for this focus becomes apparent with reading, I found the motivation for their approach therefore somewhat unclear/inconsistent.

We agree with the reviewer. The background of whole brain development and measurement of other networks are indeed to provide reference and to emphasize the extraordinarily rCBF development of the default mode network during infancy. To make the focus of the default mode network clear for the entire manuscript, we have changed manuscript title from “Infant brain regional cerebral blood flow increases supporting emergence of vital functional networks” to “Infant brain regional cerebral blood flow increases supporting emergence of the default-mode network” to emphasize the default-mode network. Please also see our response to the Essential Revisions comment #4 summarized by the editor.

2. Could the authors better describe in the mansucript why they chose to focus only on three networks rather than whole brain approaches or suggest what they might consider doing (such as linked ICA together with anatomical data) in future studies given the available data?

We thank the reviewer for this comment. These three networks have been relatively more extensively studied in the literatures (Gilmore et al., 2018; Smith et al., 2009). As the initial effort to tackle the relationship between rCBF and functional networks, we only incorporated clearly defined and extensively studied infant networks, which are the three networks described in this paper. Future studies of other infant networks including language network are warranted. In fact, we already conducted preliminary analysis investigating relationship between rCBF and functional connectivity in the language network (Ouyang et al., 2020).

Gilmore, J.H., Knickmeyer, R.C., and Gao, W. (2018). Imaging structural and functional brain development in early childhood. Nat. Rev. Neurosci. 19(3), 123-137.

Ouyang, M., Yu, Q., Kang, H., Peng, Y., Hong, B., and Huang, H. (2020). Delineation of language network maturation during infancy with multi-modal perfusion and functional MRI. Proceedings of ISMRM (Magna Cum Laude Award).

Smith, S.M., Fox, P.T., Miller, K.L., Glahn, D.C., Fox, P.M., Mackay, C.E., Filippini, N., Watkins, K.E., Toro, R., Laird, A.R., et al. (2009). Correspondence of the brain's functional architecture during activation and rest. Proc. Natl. Acad. Sci. USA 106, 13040-13045.

3. With the networks specifically, it is possible that the use of the DMN maps derived from the older infants only, may bias the results as it could then be argued that it is unsurprising that connectivity (and potentially rCBF is less) in those infants. Would it be possible to see if an age-specific representation of the network (using a method such as dual regression) would still yield the same results?

This comment was included as Essential Revisions comment #3 summarized by the editor. Please see our response to that comment. Briefly, we argued that age-specific ROIs may lead to inconsistent measurement and add artificial bias to the measurement aimed for quantifying age-related blood flow changes. We also have strong counter example showing consistent network ROIs generated from older infant data do not lead to biased higher functional connectivity in the older infants, as shown in nonsignificant age-related change of functional connectivity within the visual (Vis) and somatosensory (SM) networks in Figure 1—figure supplement 2.

4. It is mentioned in the results that the infants were sedated for study – the type of sedation should be mentioned (the drug and dose) and the possible implications on the results (especially the CBF) considered. This is also the case for the clinical history of the infants – a large proportion of them had a history of seizures, which is a concern as it questions the generalisability of the results to the healthy brain (even if they are apparently neurologically normal).

We thank the reviewer for this comment. Chloral hydrate, with a dose of 0.5 ml/kg and no more than 10 ml in total, was taken orally for each infant before scanning for sedation. Previous studies (Li et al., 2011; Suzuki et al., 2021) suggested no significant impact of chloral hydrae on CBF or sensory function. The description has been added to the *Data acquisition* subsection in Materials and Methods section.

We apologize that we stated history of seizures in error. Most infants had a simple febrile convulsion, rather than seizures with fever. These infants had no abnormality in the further neurological evaluation. Occasional simple febrile convulsion will not cause long-term brain function damage, as shown in the literature (Patterson et al., 2013; Sawires et al., 2021; Verity et al.,1992;). This statement has been corrected in the *Data acquisition* subsection in Materials and Methods section.

Li, A., Gong, L., and Xu, F. (2011). Brain-state–independent neural representation of peripheral stimulation in rat olfactory bulb. Proc. Natl. Acad. Sci. USA 108(12), 5087-5092.

Patterson, J.L., Carapetian, S.A., Hageman, J.R., and Kelley, K.R. (2013). Febrile seizures. Pediatric annals 42(12), 258-263.

Sawires, R., Buttery, J., and Fahey, M. (2021). A Review of Febrile Seizures: Recent Advances in Understanding of Febrile Seizure Pathophysiology and Commonly Implicated Viral Triggers. Frontiers in Pediatrics, 9.

Suzuki, C., Kosugi, M., and Magata, Y. (2021). Conscious rat PET imaging with soft immobilization for quantitation of brain functions: comprehensive assessment of anesthesia effects on cerebral blood flow and metabolism. EJNMMI research 11(1), 1-11.

Verity, C.M., and Golding, J. (1991). Risk of epilepsy after febrile convulsions: a national cohort study. British Medical Journal 303(6814), 1373-1376.

5. As mentioned above, it is a concern that the same acquisition and processing parameters were used for all of the infants. The T1 of arterial blood for example, seems to have been derived from published neonatal values which are likely to differ greatly from those needed for a 2 year old infant. In contrast, arterial transit time is likely to vary across development, and so the use of the same post-label delay time of 1600ms (which is markedly shorter than the value of 2000ms suggested in the Alsop white paper) for all infants is a concern – as this may lead to an underestimation of CBF in the youngest infants. Partial voluming effects will be worse in the youngest infants (especially with a slice thickness of 5mm), which could also lead to an underestimation of CBF as separation of the white matter and cortex is challenging.

We thank the reviewer for bringing up this good point. This comment has been partly included as the Essential Revisions comment #2 by the editor. Here, we would like to address the comment more systematically in the following aspects below: (1) infant physiological parameters including arterial transit times (ATT) and imaging parameter selections in the interval between neonates and adults, (2) calibration of rCBF maps with phase-contrast MRI, and (3) amelioration of partial volume effects with carefully testing multiple thresholds in the averaged gray matter probability maps.

In terms of acquisition, we used the same parameters on purpose for data harmonization. This was also a feasible way and less error-prone for MR technologists to handle the scan whenever there was infant getting recruited. As the reviewer pointed out, a post-label delay (PLD) time of 2000ms was recommended for neonates, while a PLD of 1500ms was recommended for children in the Alsop white paper. However, please note brain perfusion parameters of infants (1.37 to 24.36 months) in this study are quite different from those of neonates (usually less than 1 week or 0.25 month of age), as artery blood flow velocities increased rapidly during the first 6 months (Liu et al., 2019). Selected PLD should be larger than ATT. Mean ATT of infants were reported less than 1500ms (Varela et al., 2014) while median ATT of neonates was reported 2260ms (Kim et al. 2018). We therefore consider it valid to select PLD of 1600ms for the studied infants.

Regarding the processing parameters, we would like to point out extra phase contrast MRI was acquired and all rCBF maps of individual infants in our study have been calibrated with phase contrast whole brain cerebral blood flow by using a normalization factor. This approach will largely minimize the effect of individual variations in processing parameters such as T1 of arterial blood or labeling efficiency in the final estimation of rCBF measurement (Aslan et al., 2010).

In this study, to minimize the partial volume effect, individual rCBF map was generated in the individual space and calibrated by phase contrast MRI to minimize the individual variations of processing parameters such as T1 of arterial blood (Aslan et al., 2010). Cortical segmentation was also conducted in individual space. Then different types of images including rCBF map and gray matter segmentation probability map in the individual space were normalized into the template space. An averaged gray matter probability map was generated after inter-subject normalization. After carefully testing multiple thresholds in the averaged gray matter probability maps, 40% probability minimizing the contamination of white matter and CSF while keeping the continuity of the cortical gray matter mask across the cerebral cortex was used to generate the binary gray matter mask shown on the left panel of Author response image 3. As demonstrated in the right three panels in Author response image 3, the rCBF measure in the cortical mask in the template space is consistent across ages for accurate and reliable voxelwise comparison across age.

**Author response image 3. sa2fig3:** The gray matter mask and segmented cortical mask overlaid on rCBF map of three representative infants aged 3, 6, and 20 months in the template space. The gray matter mask on the left panel was created to minimize the contamination of white matter and CSF while keeping the continuity of the cortical gray matter mask across the cerebral cortex. The contour of the gray matter mask was highlighted with bule line.

Aslan, S., Xu, F., Wang, P.L., Uh, J., Yezhuvath, U.S., Van Osch, M., and Lu, H., (2010). Estimation of labeling efficiency in pseudocontinuous arterial spin labeling. Magnetic resonance in medicine 63(3), 765-771.

Kim, H.G., Lee, J.H., Choi, J.W., Han, M., Gho, S.M., and Moon, Y., (2018). Multidelay arterial spin-labeling MRI in neonates and infants: cerebral perfusion changes during brain maturation. American Journal of Neuroradiology 39(10), 1912-1918.

Liu, P., Qi, Y., Lin, Z., Guo, Q., Wang, X., and Lu, H., (2019). Assessment of cerebral blood flow in neonates and infants: a phase-contrast MRI study. Neuroimage 185, 926-933.

Varela, M., Petersen, E.T., Golay, X., and Hajnal, J.V., (2015). Cerebral blood flow measurements in infants using look–locker arterial spin labeling. Journal of Magnetic Resonance Imaging 41(6), 1591-1600.

6. I find the suggestion that the rCBF measures can tell us about changes in brain metabolism rather speculative – as there are no other measures of oxygen extraction for example to specifically justify this. As the data is cross-sectional, mechanistic insight such as understanding whether one precedes the other is not possible. Figure 6 for example is not justified by the data, as it is simply not possible to know from their data if the suggested physiological changes are actually taking place.

We agree with the reviewer that we should downplay previous Figure 6. Although we have made it clear it is our hypothesis that cerebral blood flow increase is associated with synaptogenesis and synaptic efficacy increase, to downplay this hypothesis that may need further justification by the data, this previous standing-alone Figure 6 has been downgraded as a supplemental figure (current Figure 5—figure supplement 3).

7. The language in parts is a little unclear and therefore could benefit from clarification. Example include: line 53 "increasing feed of energy to fuel the brain development" or line 56 "impeding fundamental view of energy expenditure across functional systems of early developing brain"

We thank the reviewer for this comment. In the first paragraph of INTRODUCTION section, the words ‘increasing feed of energy to fuel the brain development’ has been changed to ‘increasing energy consumption of the brain’. The words ‘impeding fundamental view of energy expenditure across functional systems of early developing brain’ have been changed to ‘impeding understanding of energy expenditure across functional systems of early developing brain’.

Specific points:1. I would prefer rewording of the title if possible – the use of the word "vital" does not seem appropriate, as this is not demonstrated in the work.

As suggested, the word ‘vital’ has been removed from the title and abstract.

2. In the first line of the abstract, the sentence reads rapid cerebral blood flow (rCBF) which incorrectly implies that this is what the acronym stands for.

We thank the reviewer for catching this typo. ‘most rapid cerebral blood flow (rCBF)’ has been revised to ‘most rapid regional cerebral blood flow (rCBF)’ in the abstract.

3. The abstract mentions "rCBF dynamics" – which I would caution against. In this work, they are really only looking at the mean rCBF across the acquisition, rather than its specific dynamics across the volumes acquired.

‘rCBF dynamics’ has been changed to ‘rCBF increase’ in the abstract. We have also changed ‘dynamics’ to ‘increases’ or ‘changes’ in the title and across the text to be consistent.

4. Suggest rewording of line 30 "We found faster rCBF increases in the DMN than other regions." as whilst this is true in comparison to the visual/SM networks – it wasn't demonstrated in comparison to all other regions?

As suggested, ‘other regions’ has been replaced by more specific ‘visual and sensorimotor networks’ in the abstract.

5. The statement in line 122 of "high quality of the rCBF maps" is not justified as its not clear how this "quality" has been defined – suggest rephrasing, please.

We thank the reviewer for this comment. ‘High quality of the rCBF maps’ has been rephased as ‘The rCBF maps with high gray/white matter contrasts’.

6. The values of rCBF on line 141 should state they are the mean across all of the infants and should include some estimate of the variance.

We thank the reviewer for this comment. Standard errors have been added in the subsection *Faster rCBF increases in the DMN hub regions during infant brain development* in the Results section to clarify the mean and stand errors in those rCBF measurements.

7. I note that rCBF is consistently higher at the base of the brain and occipital regions – is this a recognized pattern in the literature? If not, could it be because vascular crushers were not used and there is overestimation due to the residual blood in the basilar artery?

We thank the reviewer for bringing up this good point. This is a recognized pattern in the literature. This comment has been included as the Essential Revisions comment #1 by the editor. Please see our response to that comment. Relevant literature is also listed in the response to Essential Revisions comment #1.

Reviewer #3 (Recommendations for the authors):The manuscript describes a study of 48 infants aged 0-24 months who underwent resting state functional MRI (rs-fcMRI) and arterial spin labeling regional blood flow (rCBF) measurements. The authors focused on the default mode network (DMN), visual cortex (VIS), and somatomotor cortex (SM). They found faster increases in rCBF in the DMN with strongly coupled increases of rCBF to network "strength" measured through rs-fcMRI. They concluded that the faster increase in rCBF in the DMN takes place to meet the metabolic demands of DMN maturation.The authors found stronger correlations between increases in rCBF and rs-fcMRI in the DMN than in VIS and SM. They suggest that this indicates that changes in rCBF constitute a physiological underpinning of DMN maturation. However, it is not clear that the underlying assertion that increases in rs-fcMRI (developmental maturation) and increases in rCBF are tightly linked is supported by the data.First, VIS and SM are both more mature than DMN at birth, yet VIS has the highest rCBF and SM has the lowest. If rCBF and maturity were tightly coupled, one would expect both to have relatively high rCBF values.

We thank the reviewer for this comment. We would like to clarify the finding is rCBF increase in a developmental period (not rCBF at baseline) is tightly coupled with maturity. Such rCBF increase is associated with the increase of energy consumption during brain development. So our finding is not contradictory to the observation that Vis has the highest rCBF and SM has the lowest. We would also note this observation of the rCBF distribution pattern of relatively higher rCBF in the VIS regions and lower rCBF int eh SM regions at a certain infant time point is consistent to previous PET study (Chugani et al., 1987) and ASL study (Wang et al., 2008).

Chugani, H.T., Phelps, M.E., and Mazziotta, J. C. (1987). Positron emission tomography study of human brain functional development. Annals of neurology 22(4), 487-497.

Wang, Z., Fernández-Seara, M., Alsop, D.C., Liu, W.C., Flax, J.F., Benasich, A.A., and Detre, J.A., (2008). Assessment of functional development in normal infant brain using arterial spin labeled perfusion MRI. Neuroimage 39(3), 973-978.

Second, the data shown in Figure 2A show that while rCBF increases with maturation, it does so along posterior-to-anterior and inferior-to-superior gradients which do not appear to correlate with regional variability in maturation. The SM region chosen for the study is located mainly in the leg area, which is primarily in the superior and medial SM cortex. The more inferior SM areas, corresponding to the arm and face, are not included. Given an inferior-to-superior gradient in the maturational increase in rCBF, the more superior portion of SM chosen is relatively late to show an increase in rCBF.

We thank the reviewer for this comment. We believe the maturation pattern is more complicated than relatively simplified posterior-to-anterior and inferior-to-superior gradients. It also presents a primary-to-association gradient reproducibly found in the literature (see Sydnor et al., 2021 for review). Such mixed patterns are consistent with the maturation pattern demonstrated in current Figure 3a (previous Figure 2a). More clear maturation gradient could benefit from future rCBF maps of higher signal-to-noise ratio and higher resolution.

The reviewer brought up an interesting point of incomplete SM areas. As stated in the manuscript, all ROIs including SM areas were data driven instead of mapped from SM from an atlas. Thus lack a portion of the functional area could inevitably happen. Using independent component analysis (ICA), we did find another ROI that might be related to inferior SM region, as shown in Author response image 4. However, this ROI clearly included auditory cortex. To avoid confounding factor from other functional areas, this ROI was not included as the SM region. In Author response image 4, we demonstrated although this ROI is located inferiorly, the maturation pattern of this ROI in terms of rCBF increase rate is not significantly different (p>0.05) from that of the SM included in the manuscript (Figure 1—figure supplement 1b) and located more superiorly.

Sydnor, V.J., Larsen, B., Bassett, D.S., Alexander-Bloch, A., Fair, D.A., Liston, C., Mackey, A.P., Milham, M.P., Pines, A., Roalf, D.R., et al., (2021). Neurodevelopment of the association cortices: Patterns, mechanisms, and implications for psychopathology. Neuron 109, 2820-2846.

**Author response image 4. sa2fig4:** (a) An ROI revealed by ICA analysis included inferior SM and auditory cortex. (b) In ROI shown in a, no significant difference (p > 0.05) of rCBF increase rate between this ROI and SM network ROI included in the manuscript was found. The rCBF of this ROI also increases significantly with age (r = 0.62, p<10^-4^).

Third, there are fairly large differences in the rCBF of subregions of the DMN, though one would expect them to be similar if they are maturing as a unit.

The DMN subregions identified with data driven approach from resting-state functional MRI (rs-fMRI) data and shown in current Figure 1—figure supplement 1 are consistent to the DMN subregions reproducibly identified by many other studies (see Raichle 2015 for review). The synchronized blood oxygenation level dependent (BOLD) signal from rs-fMRI BOLD has identified these subregions as a unit. Furthermore, despite various rCBF baseline values in these DMN subregions, the rCBF increase rates in these DMN subregions are consistent (Figure 3c). Despite that these DMN subregions are anatomically separate, it is reasonable to consider they are maturing functionally and physiologically together as a unit.

Raichle, M.E. (2015). The brain's default mode network. Annu. Rev. Neurosci. 38, 433-447.

Finally, the stronger correlation between increases in rCBF and rs-fcMRI metrics for the DMN may reflect its greater increase in rCBF during the time period studied. This may be due to the finding that VIS, located posteriorly and inferiorly, has already undergone its greatest increase in rCBF; while SM, located superiorly, has yet to undergo its greatest increase in rCBF. Thus both have relatively modest changes in rCBF during the period studied. The steeper rise in rCBF in the DMN, which may be by virtue of its location, would allow for a stronger correlation with maturational increases in rs-fcMRI parameters.

We thank the reviewer for this comment. Before original submission, we too were suspecting the stronger correlation between rCBF and rs-fcMRI metrics for the DMN only reflects greater increase in rCBF in the DMN. We added permutation tests with 10,000 permutations exactly to rule out that coupling in the DMN ROIs was driven by greater increase in rCBF in those ROIs. The results shown in Figure 5 clearly demonstrated strong coupling only localized at in the DMN regions. Figure 5—figure supplement 2 provided counter example that despite greater increase in rCBF in the DMN regions, rCBF increase in the DMN is not coupled to the rs-fcMRI metrics of the VIS or SM networks. We believe the combined Figure 5 and Figure 5—figure supplement 2 are robust to demonstrate strong coupling is specifically localized in the DMN regions, but not SM or VIS regions. Please also see related response to R1-2.

We agree with the reviewer that the VIS has already undergone its greatest increase in rCBF before the studied infant development period focused in this manuscript. Our previous study (Cao et al., 2017) on rs-fcMRI suggested SM located both inferiorly and superiorly also has already undergone its greatest functional connectivity increase during perinatal development, before the infant developmental stage focused in this manuscript.

Cao, M., He, Y., Dai, Z., Liao, X., Jeon, T., Ouyang, M., Chalak, L., Bi, Y., Rollins, N., Dong, Q., et al. (2017). Early development of functional network segregation revealed by connectomic analysis of the preterm human brain. Cereb. Cortex. 27, 1949-1963.

Since the gradient in increasing rCBF during development appears to be based on relative location, it is important to consider whether the superior-inferior gradient might be a technical artifact related to applying labeling pulses to the neck, which is getting longer, or changes in flow velocities with age.

We thank the reviewer for bringing up this good point. It is unlikely that the superior-inferior is a technical artifact. This comment has been included as the Essential Revisions comment #1 by the editor. Please see our response to that comment.

[Editors’ note: further revisions were suggested prior to acceptance, as described below.]

Reviewer #1 (Recommendations for the authors):The authors have partly responded to the reviewers' comments in my opinion. Some points still seem to me to be questionable in the manuscript, whereas some answers provided by the authors in their replies would have benefited from being added to the manuscript.

We agree with the reviewer’s comments. We did not put all responses into the manuscript since in *eLife* response letters are also published along with the manuscript. But in this revision, following this reviewer’s suggestion, these responses have now been added to the manuscript. Please see below for details.

– Essential revisions 1 and 2: unless I am mistaken, these elements have not been added to the manuscript. Perhaps this would be relevant for readers who don't read the reviewers' comments and the authors' answers in addition to the article.

We thank the reviewer for this comment. We have added our responses to previous essential revisions 1 and 2 to the Discussion section.

– R1-1: The answer provided might seem insufficient. The grey matter masks shown seem to me to clearly contain white matter, at least in the youngest infants for whom the cortex is very thin. The problem of reliably identifying the cortex to limit partial volume effects is definitely complex, and pCASL acquisition with thick slices doesn't avoid this problem anyway. That said, I still trust the measures performed, but I think it is important to discuss this point so that the reader is aware of this potential measurement bias. One way of doing this would be to estimate what the bias might be as a function of age (based on the estimated proportions of grey/white matter in the masks considered, and estimated rCBF in each tissue).

We agree with the reviewer that partial volume effects are a consideration factor for rCBF maps especially for smaller brain and thinner cortex of younger infants. Elaborated in our response to R1-1 in the last revision, we have adopted a method making the rCBF measure in the cortical mask in the template space consistent across ages for accurate and reliable voxel-wise comparison across age. As this reviewer suggested, the partial volume effect could be heterogenous to different ages with thinner cortex of younger infants. To address that, *individual phase contrast (PC) MRI of all infants* were obtained in our study to calibrate all rCBF maps of individual infants obtained with ASL scans by applying a normalization factor. In this way, the possible underestimation of rCBF in the younger infants due to more severe partial volume effects can be corrected. A function of age suggested by this reviewer to calibrate partial volume effects could also be effective but might require gold-standard infant gray matter rCBF measurements at different age as well as gold-standard segmentation of gray and white matter of infants. The former data is what we aim to obtain. The latter is known to be difficult and lacks consensus especially for younger infants due to poor T1-weighted or T2-weighted contrasts. Relevant discussion has been added in the last paragraph of the Discussion section. Please also see our response to RR2-3.

– R1-2: Again, the answer provided seems incomplete. When two measures depend on the same variable (age), their correlation can be partly driven by the common correlation, and it cannot then be considered as specific. The similarity between figures 3b and 5d on the lateral side draws attention to this point. Unless I am mistaken, the measures of functional connectivity in the visual and sensorimotor networks are much less age-dependent than in the DMN network (Figure 1-sup2), whereas the rCBF is age-dependent in all regions: this difference might drive the difference in FC-rCBF correlation observed across networks. A simple way to "prove" that the FC-rCBF correlation is indeed specific for the DMN would be to perform all these correlation analyses after removing all age effects in all measurements (region-wise or vertex-wise). These results would be convincing.

We fully understand the reviewer’s concern, as this was exactly our concern which motivated us to use the permutation tests to test whether the functional connectivity (FC)-rCBF correlation is due to their joint dependence on age. This permutation tests resulting in Figure 5 were elaborated in our response to R1-2 in last revision. Following the suggestion in this comment, we conducted further analysis testing the FC-rCBF correlation in the DMN regions after removing all age effects and found that, after regressing out the age effects, the correlation remains significant in both 0-12-month (r = 0.303, p < 0.001) and 12-24-month (r = 0.217, p < 0.001) groups. This result, combined with Figure 5, convincingly indicates that our finding of coupling between FC and rCBF cannot be explained just by age-related increases of both FC and rCBF in the DMN regions. The new result has been added in the subsection *Coupling between rCBF and FC within DMN during infant brain development* in the Results section.

– R1-4: it seems to me that the order of the method paragraphs should be changed to follow the new order of the results.

We thank the reviewer for this comment. The order of the paragraphs has been reorganized accordingly in the Materials and Methods section.

– R1-11: citation of the Cusack et al. paper would be welcome in the manuscript.

As suggested, citation of the Cusack et al. paper has been added to the subsection *Data acquisition* in Materials and Methods section.

– Introduction p3: the sentence "there have been no known whole-brain mappings of heterogeneous infant brain regional cerebral blood flow (rCBF) changes across landmark infant ages thus far" is incorrect (see the Lemaitre et al. paper pointed out in the previous review).

We thank the reviewer for this comment. We have revised the sentence in the Introduction section.

Reviewer #2 (Recommendations for the authors):Thank you for the detailed answers to the questions and amendments to the manuscript. Whilst I agree with some of their responses, I feel there are still some unresolved issues which I think are fundamental to resolve:

We thank the reviewer for this comment. It seems some remaining issues are due to lack of clarification or miscommunication (e.g. same post-label delay (PLD) instead of misunderstood different PLD). As can be seen from our responses to these remaining comments, we explicitly acknowledged limitations of relatively lower pCASL resolution related to partial volume effects. All our imaging parameters were carefully selected and well justified. More importantly, for measurement accuracy all infant pCASL rCBF maps were calibrated by extra acquired subject-specific phase contrast MRI. Please see our responses to individual points below.

1. The use of similar acquisition parameters for all age groups: as far I understand it, the method proposed by Aslan et al. using PCA to calculate the labelling efficiency still requires the T1 of arterial blood as a parameter – for which neonatal values (according to their references) have been used. It would be good to know what would happen if more developmentally appropriate values were used in their CBF estimations. Secondly, the Aslan method provides a single corrective factor (and estimate of the labelling efficiency) for the CBF estimation based on the PCA measurement (which is global). Therefore it cannot correct for any regional effects of using a different post-label delay in the ASL acquisition – with knock-on effects for the regional CBF estimation (such as the spatial bias leading to the inferior-superior gradient for example). Would it be possible for the authors to provide some empirical evidence that the post-label delay has not influenced their results?

We thank the reviewer for this comment. We would like to point out that according to the rCBF calculation equation in the ASL white paper (Alsop et al., 2015), similar to the effect of labeling efficiency parameter on the rCBF map, the effect of arterial blood T1 on rCBF map estimation is global. By making averaged rCBF from pCASL equal to global CBF obtained from extra acquired phase contrast MRI of each infant using a "subject-specific" normalizing factor, we calibrated pCASL rCBF maps of all individual infants in this study. With this “subject-specific” calibration approach, the effects of individual variations in processing parameters such as T1 of arterial blood or labeling efficiency in the final estimation of rCBF measurement were largely minimized. However, for studies without extra phase contrast MRI acquired for "subject-specific" rCBF calibration, utilizing developmentally appropriate (as suggested by this reviewer) or even subject-specific T1 values of arterial blood is encouraged. We added discussion on the effect of T1 of arterial blood in rCBF estimation in the last paragraph of the Discussion section.

Secondly, according to the ASL white paper (Alsop et al. 2015), selected PLD should be larger than the arterial transit time (ATT) for reducing regional CBF bias. In this study by using a consistent PLD of 1600 ms larger than ATT of infants, the effect of PLD on rCBF differences across subjects or brain regions is relatively trivial. Please note the mean ATT of infants was reported to be less than 1500 ms (Varela et al. 2015; Kim et al. 2018). We also respectfully point out that we used in ASL acquisition a consistent PLD instead of a different PLD indicated in this comment. We acknowledge that due to lack of consensus on ASL acquisition parameters for the infant population, further studies focusing on systematically optimizing ASL acquisition protocol in the infant population are needed. Relevant details and discussion have been added to the subsection *Data acquisition* in Materials and Methods section.

Alsop, D.C., Detre, J.A., Golay, X., Günther, M., Hendrikse, J., Hernandez‐Garcia, L., Lu, H., MacIntosh, B.J., Parkes, L.M., Smits, M., et al. (2015). Recommended implementation of arterial spin‐labeled perfusion MRI for clinical applications: A consensus of the ISMRM perfusion study group and the European consortium for ASL in dementia. Magn Reson Med. 73, 102-116.

Kim, H.G., Lee, J.H., Choi, J.W., Han, M., Gho, S.M., and Moon, Y., (2018). Multidelay arterial spin-labeling MRI in neonates and infants: cerebral perfusion changes during brain maturation. American Journal of Neuroradiology 39, 1912-1918.

Varela, M., Petersen, E.T., Golay, X., and Hajnal, J.V., (2015). Cerebral blood flow measurements in infants using look–locker arterial spin labeling. Journal of Magnetic Resonance Imaging 41, 1591-1600.

2. With regards to the inferior-superior gradient, the references that the authors cite do not show the same kind of effects. Whilst many of them do suggest that CBF is higher posteriorly and is higher in the occipital lobe – it is not a "hot spot" in the same way as seen in the maps here (especially at the youngest ages). The maps in the Lamaitre work for example look very different (see figure 2 in their paper) to those shown in figure 3 here.

We agree with the reviewer that rCBF increase pattern is more complicated than relatively simplified inferior-to-superior or posterior-to-anterior gradient. Our Figure 3 suggests that rCBF increase pattern also presents a primary-to-association gradient reproducibly found in the literature (see Sydnor et al., 2021 for review). The differences between our Figure 3 with Figure 2 of Lemaitre Cereb Cortex 2021 might result from the two factors. First, the age range is different. In the same age range from 0 to 12 months, the rCBF maps are highly similar. The 12-month rCBF map in Lamaitre study also presents relatively higher rCBF in the occipital lobe, lateral temporal cortex, and lateral/medial prefrontal cortex. The “hot spot” appearance is more apparent in the 18-month and 24-month rCBF maps which are not available in Lemaitre study and are therefore not comparable. Second, although the pCASL perfusion MRI acquisition protocol we adopted in this manuscript is state-of-the-art, the voxel size is slightly larger than that of Lemaitre study. We believe slightly larger voxel size may contribute to blurring the rCBF map resulting in a “hot spot” in the inferior part of the brain. Using larger voxel size was by our design to ensure sufficient signal-to-noise-ratio (SNR) of the rCBF map. A 3T scanner used in this study rather than a 1.5T scanner in Lemaitre study also helps improve SNR for more accurate rCBF measurement, as seen in Figure 2 in this manuscript. Future studies with higher resolution and higher SNR are warranted to delineate the finer details of rCBF increase pattern in infants. The last paragraph in the Discussion section has been revised accordingly.

Sydnor, V.J., Larsen, B., Bassett, D.S., Alexander-Bloch, A., Fair, D.A., Liston, C., Mackey, A.P., Milham, M.P., Pines, A., Roalf, D.R., et al., (2021). Neurodevelopment of the association cortices: Patterns, mechanisms, and implications for psychopathology. Neuron 109, 2820-2846.

3. I am unconvinced by the argument that calculating grey matter maps in native high-resolution space and registering them together with the low-resolution CBF map into standard space thresholded probabilistic maps resolve the partial voluming issue. Fundamentally this does not change the base acquisition voxel which is very large and is the same across all of the infants studied – so differences relative to the size of the brain/cortex and partial voluming in the acquisition itself in the acquisition voxel are surely still an issue?

We thank the reviewer for this comment. We agree with the reviewer that partial volume effects of relatively lower resolution pCASL acquisition could be heterogeneous for infants at different ages as smaller brain and thinner cortex of younger infants cause larger partial volume effects when the imaging parameters are consistent across infants. Please note it is not the process of low-resolution rCBF map registered into standard space thresholded probabilistic maps that resolves the partial voluming issue. Instead, it is calibration with *individual phase contrast (PC) MRI* that contributes to correcting for bias of heterogenous partial volume effects. Please see details in our response to RR1-2. We also explicitly acknowledged heterogenous partial volume effects in the last paragraph of the Discussion section.